# Advances in Diagnostic Tools and Therapeutic Approaches for Gliomas: A Comprehensive Review

**DOI:** 10.3390/s23249842

**Published:** 2023-12-15

**Authors:** Gayathree Thenuwara, James Curtin, Furong Tian

**Affiliations:** 1School of Food Science and Environmental Health, Technological University Dublin, Grangegorman Lower, D07 H6K8 Dublin, Ireland; gayathreethenuwara57@gmail.com; 2Institute of Biochemistry, Molecular Biology, and Biotechnology, University of Colombo, Colombo 00300, Sri Lanka; 3Faculty of Engineering and Built Environment, Technological University Dublin, Bolton Street, D01 K822 Dublin, Ireland; james.curtin@tudublin.ie

**Keywords:** gliomas, multimodal therapeutic strategies, precision medicine, diagnostic tools, brain tumor therapeutics

## Abstract

Gliomas, a prevalent category of primary malignant brain tumors, pose formidable clinical challenges due to their invasive nature and limited treatment options. The current therapeutic landscape for gliomas is constrained by a “one-size-fits-all” paradigm, significantly restricting treatment efficacy. Despite the implementation of multimodal therapeutic strategies, survival rates remain disheartening. The conventional treatment approach, involving surgical resection, radiation, and chemotherapy, grapples with substantial limitations, particularly in addressing the invasive nature of gliomas. Conventional diagnostic tools, including computed tomography (CT), magnetic resonance imaging (MRI), and positron emission tomography (PET), play pivotal roles in outlining tumor characteristics. However, they face limitations, such as poor biological specificity and challenges in distinguishing active tumor regions. The ongoing development of diagnostic tools and therapeutic approaches represents a multifaceted and promising frontier in the battle against this challenging brain tumor. The aim of this comprehensive review is to address recent advances in diagnostic tools and therapeutic approaches for gliomas. These innovations aim to minimize invasiveness while enabling the precise, multimodal targeting of localized gliomas. Researchers are actively developing new diagnostic tools, such as colorimetric techniques, electrochemical biosensors, optical coherence tomography, reflectometric interference spectroscopy, surface-enhanced Raman spectroscopy, and optical biosensors. These tools aim to regulate tumor progression and develop precise treatment methods for gliomas. Recent technological advancements, coupled with bioelectronic sensors, open avenues for new therapeutic modalities, minimizing invasiveness and enabling multimodal targeting with unprecedented precision. The next generation of multimodal therapeutic strategies holds potential for precision medicine, aiding the early detection and effective management of solid brain tumors. These innovations offer promise in adopting precision medicine methodologies, enabling early disease detection, and improving solid brain tumor management. This review comprehensively recognizes the critical role of pioneering therapeutic interventions, holding significant potential to revolutionize brain tumor therapeutics.

## 1. Introduction

Gliomas represent a highly prevalent and formidable category of primary brain tumors, characterized by a grim prognosis attributable to their invasive potential and aggressive clinical behavior. These tumors occupy a significant share of primary brain malignancies, accounting for more than 80% of cases and constituting approximately 30% of all brain tumors [1]. The recent update by the World Health Organization (WHO) expanded the brain tumor classification by integrating genotypic markers alongside the previously considered histological markers. At present, the glioblastoma classification involves identifying a specific single nucleotide polymorphism in the isocitrate dehydrogenase (IDH) gene, distinguishing between wild-type and mutant variations. Previously, the diagnosis of glioblastomas relied on histological features, like microvascular proliferation or necrosis, encompassing both IDH-mutated (10%) and IDH wild-type (90%) tumors, each displaying markedly distinct biological natures and prognoses. However, under WHO CNS5, glioblastomas are at present exclusively attributed to IDH wild-type tumors, marking a substantial departure from the previous classification system. Furthermore, within the updated classification, IDH wild-type diffuse astrocytic tumors in adults, which lack the typical histological features of glioblastoma but exhibit any of three specific genetic parameters (TERT promoter mutation, EGFR gene amplification, or the combined gain of the entire chromosome 7 and the loss of the entire chromosome 10 [+7/−10]), also fall under the category of glioblastomas. Conversely, all IDH-mutant diffuse astrocytic tumors are consolidated in a single category (astrocytoma, IDH-mutant) and graded as 2, 3, or 4 in this new classification system [2]. Despite the implementation of multimodal therapeutic strategies, the median overall survival for glioblastoma patients remains around 14 to 15 months. The alignment between these time frames emphasizes the critical need for the development of more potent treatment modalities to notably improve the survival outcomes of individuals facing glioblastoma [3,4].

The current therapeutic approach for gliomas involves a multimodal strategy, typically commencing with extensive surgical resection, followed by radiation and chemotherapy [5,6]. However, these tactics often fall short of achieving the desired clinical outcomes due to high recurrence rates and the gradual development of drug resistance over time. Glioma treatment is further complicated by several factors. Firstly, gliomas exhibit high infiltration, making complete cellular-level resection nearly impossible [7]. They contain hypoxic regions that provide niches for glioma-initiating cells, which can yield more aggressive recurrent tumors that are resistant to radiation and chemotherapy [8,9]. Secondly, the large intertumor and intratumor heterogeneity hinders the development of targeted therapies [10]. Various genetic and epigenetic markers have led to diverse classification systems, and recent studies have shown spatial and temporal variations within the same tumor [11]. Thirdly, the blood–brain barrier (BBB) restricts the delivery of chemotherapeutic drugs to the brain. The poorly formed, leaky blood vessels exhibit enhanced permeability but are not uniform throughout the tumor [12,13]. Furthermore, efflux pumps upregulated by glioblastoma cells limit drug penetration into tumor cells [14,15]. Lastly, the immunosuppressive microenvironment within it presents a challenge. Some lack pre-existing tumor T-cell infiltration, making them resistant to immune checkpoint inhibitors (Figure 1) [10,11,12,13,14,15]. These tumors have defects in antigen presentation and accumulate immunosuppressive cells, limiting the efficacy of immunotherapies [16,17]. Overcoming these challenges in glioma treatment necessitates innovative and combinatorial approaches to improve patient outcomes.

Conventional diagnostic methods for identifying gliomas represent the initial steps in detecting these primary brain tumors. Computed tomography (CT) scans and magnetic resonance imaging (MRI) are among the standard imaging techniques used to provide a foundational understanding of the tumor’s location, size, and characteristics. These essential diagnostic tools serve as the entry point for a more comprehensive assessment and management of gliomas. In addition to CT scans and MRI, positron emission tomography (PET) scans are another conventional imaging method used in the diagnosis of gliomas.

The integration of contrast-agent-enhanced CT marked a significant milestone in contemporary neuroimaging, enabling the precise anatomical localization of brain tumors, particularly the malignant ones due to the enhanced contrast it offers [18]. CT stands out with its widespread availability, faster scanning times, and lower cost compared to MRI [19]. However, it is important to consider that CT exposes patients to radiation, and this exposure can accumulate when repeated imaging is necessary. Additionally, the CT’s ability to visualize soft tissues is notably inferior to that of MRI, which offers a higher resolution [20].

The development of MRI diffusion-weighted sequences has been transformative in neuroimaging, allowing for an indirect estimation of tumor cellularity. This innovation has significantly replaced CT in the diagnosis of glioblastomas, highlighting the evolving landscape of diagnostic methods [19].

MRI is the preferred imaging modality for diagnosing and characterizing glioblastomas due to its high sensitivity to the tumor presence and associated features, including peritumoral edema. This lesion, known for its infiltrative nature, often extends beyond the visible margins of the abnormal signal intensity on MRI scans. While a formal diagnosis of glioblastomas requires histopathology and genetic markers, structural MRI scans are routinely conducted to aid in surgical guidance [20,21].

MRI, using various sequences, such as T1-weighted, T2-weighted, and gadolinium-enhanced, plays a crucial role in diagnosing, characterizing, monitoring, and assessing the treatment of gliomas [22]. It excels in providing high-resolution structural details, offering valuable insights into the tumor location and size. However, one notable limitation of the conventional MRI lies in its lack of biological specificity [23]. For instance, T2-weighted signals primarily reflect tissue water content, while contrast enhancement signifies increased blood–brain barrier permeability. These factors make it challenging to non-invasively diagnose and accurately characterize gliomas. Furthermore, distinguishing active tumor regions from treatment-related effects proves intricate, exacerbated by the complex and subtle morphological changes in gliomas that are often imperceptible to the naked eye, even for experienced radiologists [22,23]. The commonly employed response criteria, relying on linear measurements of enhancing tumor components, encounter difficulties due to the irregular shape and heterogeneous composition of gliomas, leading to poor correlations with clinical outcomes. While the conventional MRI is widely available and delivers essential anatomical information, the absence of pathology-specific biomarkers and limitations in image analysis methodologies hamper its diagnostic and prognostic efficacy [22].

Positron emission tomography (PET) has emerged as a pivotal tool in the diagnosis, prognosis, and monitoring of glioblastomas. It offers insights beyond what magnetic resonance imaging (MRI) can provide, delving deeper into the biological aspects of these brain tumors. This additional information proves invaluable for non-invasive grading, differential diagnosis, outlining tumor extent, surgical planning, radiotherapy, and post-treatment monitoring. In clinical applications, two primary classes of radiotracers are predominantly used for imaging glioblastomas: those related to glucose metabolism and those related to amino acid transport [24]. Both classes of tracers offer valuable insights into glioma grading and prognosis. The amino acid tracers O-(2-18F-fluoroethyl)-L-tyrosine (FET) (18F-FET), carbon-11-methyl-L-methionine (MET) (11C-MET), 3,4-dihydroxy-6-18F-fluoro-L-phenylalanine (FDOPA) (18F-FDOPA), α-[11C] methyl-l-tryptophan (AMT) (11C-AMT), and 18F-fluciclovine (18F-FACBC) exhibit a lower uptake in the normal brain tissue and excel in aiding to delineate the tumor extent, design treatment strategies, and facilitate follow-up. Their main attribute lies in creating high contrast between malignant tissues and the normal brain tissue by exhibiting a reduced uptake in the latter. This capability outperforms the abilities of 18F-2-fluoro-2-deoxy-D-glucose (18F-FDG) in brain tumor imaging [24,25]. Recent advancements in PET imaging using radiolabeled amino acids have been transformative. These efforts have prompted the international Response Assessment in Neuro-Oncology (RANO) working group to recommend amino acid PET as an essential additional tool in the diagnostic assessment of brain tumors [26]. This recognition underscores the growing importance of PET in advancing the understanding and management of glioblastomas. As this technology continues to evolve, it promises to play an increasingly pivotal role in the battle against these formidable brain tumors. This recognition underscores the growing importance of PET in advancing the understanding and management of glioblastomas.

In parallel, clinical trials play a pivotal role as essential scientific investigations, significantly contributing to the progress in comprehending and treating glioblastomas. These trials serve as dynamic platforms for testing innovative therapies, exploring novel treatment modalities, and assessing the efficacy and safety of emerging interventions. The integration of novel theragnostic approaches into clinical trials for glioblastomas represents a groundbreaking endeavor aimed at transforming the landscape of treatment strategies for this formidable brain tumor. The imperative for such transformative approaches is underscored by the existing limitations in the current therapeutic and diagnostic methods, emphasizing the urgent need for the development of safer, more efficient, and highly targeted theragnostics for individuals affected by brain cancer. Diligent research efforts are underway to unearth innovative approaches that can effectively identify tumors, regulate tumor progression, and address the issues of drug resistance and tumor recurrence. The current glioma treatment paradigm, characterized by a “one-size-fits-all” approach, demonstrates its limitations in achieving meaningful outcomes. In this context, we enumerate nine clinical trials conducted for the treatment of glioblastomas since the year 2009. The clinical trials were listed with the following parameters: treatment, phase, concentration/dose, sample size, result, country, and year [27,28,29,30,31,32,33,34,35,36] (Table 1). Figure 2 illustrates the nations where the clinical trials were conducted for glioblastoma treatment.

The different combinations of lomustine (CCNU), temozolomide (TMZ), nivolumab monotherapy (NIVO), ipilimumab (IPI), mebendazole (MBZ), dabrafenib, trametinib, selinexor, herpes virus G47∆, and ultrasound have been employed in clinical trials [27,28,29,30,31,32,33,34,35,36] (Table 1). The clinical trials for glioblastoma treatment were conducted in the USA, Canada, EU, Australia, India, Republic of Korea, and Japan. Recent advancements in technology, particularly when integrated with cutting-edge biocompatible interfaces, are promising in redefining the landscape of glioma therapeutics. These innovations have the potential to significantly reduce invasiveness while facilitating the precise, multi-pronged targeting of localized gliomas, ultimately achieving an unprecedented level of precision in treatment. As research in this field progresses, these advancements are set to offer new hope to patients grappling with the formidable challenges posed by gliomas. In the following sections, the advanced theragnostic techniques that could be used to address these pressing issues are explored.

## 2. Diagnostic Tools

### 2.1. Colorimetric Technique for Brain Cancer Diagnostic: Tumor Markers

Cancer cells and other resident non-malignant cells possess the ability to release distinct proteins referred to as “tumor markers” into the bloodstream as the cancer advances. These tumor markers are detectable in diverse sample types, including blood, urine, or tissue, and their concentrations are frequently aligned with the cancer’s stage [37]. As proteomic technologies have evolved, a plethora of protein-based tumor markers has been identified for various cancer types, underscoring their critical significance and their potential role in early cancer detection [38].

Gliomas are highly heterogeneous brain tumors, and various molecular biomarkers have been extensively studied for their diagnostic, predictive, and prognostic potential. Several major molecular biomarkers have garnered significant attention in the last five years. IDH (isocitrate dehydrogenase) mutations are central to glioma diagnosis and prognosis, with recent developments having been achieved in diagnostic methods [39]. MGMT (O6-methylguanine-DNA methyltransferase) has received considerable attention, with the DNA methylation status serving as a crucial indicator of its activity, determined through techniques such as pyrosequencing [40] and immunohistochemistry [41]. Telomerase reverse transcriptase (TERT) promoter mutations are also under scrutiny for their role in telomere maintenance in gliomas [42].

Other promising molecular biomarkers are B7-H3, chondroitin sulfate proteoglycan-4 (CSPG4), carbonic anhydrase-IX (CAIX), GD2, human epidermal growth factor receptor 2 (HER2), interleukin 13 receptor alpha 2 (IL13R α2), matrix metalloproteinase (MMP2), trophoblast-cell-surface antigen 2 (TROP2) and 1p/19q co-deletion [43,44], ATRX mutations [45], EGFR (epidermal growth factor receptor) alterations [46], CD70, CD147, CDKN2A deletions [47], exosomes [48], cfDNA (cell-free DNA) [49], ctDNA (circulating tumor DNA) [50], and CTCs (circulating tumor cells) [51]. These markers provide insights into the genetic and molecular characteristics of gliomas, influencing diagnosis and prognosis. A summary of the glioma tumor microenvironment and biomarkers is provided in Figure 3.

The development of immunoassays for tumor marker analysis has received research attention, with a variety of techniques and methods being employed for their creation. Among these techniques, colorimetric methods, which rely on visual color changes in the reaction medium, have emerged as particularly convenient and accessible [52]. Notably, they are an area of significant progress in the design of sensing systems for tumor marker detection. However, the practical application of these markers still faces sensitivity and clinical implementation challenges. Further research is undergoing to refine their utility in understanding and managing gliomas.

Colorimetric biosensors have gained prominence in various applications due to their simplicity, cost-effectiveness, and user-friendly nature. These biosensors function based on color changes triggered by different mechanisms, including the oxidation of peroxidase or peroxidase-mimicking nanomaterials, the agglomeration of nanomaterials, or the use of dye indicators. In the context of brain cancer diagnosis, colorimetric techniques have found utility in detecting tumor markers, which are molecules or biomarkers that signify the presence of cancer cells. Despite their advantages, colorimetric methods can sometimes lack the desired selectivity and sensitivity, leading to heterogeneous signals that may be misinterpreted [53]. Catalytic reactions and enzymatic conversions are common strategies in colorimetric sensing, where enzymes, like peroxidase, catalyze the oxidation of substrates to produce color changes. However, enzymes have limitations, prompting the development of catalytic nanomaterials, such as metal nanoclusters, which can mimic peroxidase’s activity and enable the colorimetric detection of cancer cells, among other applications [54]. Gold nanoparticles (GNPs) have received attention due to their unique optical properties, particularly their capacity to exhibit color changes in response to alterations in their local environment [55]. Engineered to bind selectively to glioma-associated biomarkers, functionalized GNPs enable the specific detection of these markers. When the binding occurs, the GNPs aggregate, leading to a shift in their plasmon resonance frequency, subsequently resulting in a visible change in color. The discernible color shift can be detected and quantified using straightforward spectrophotometric or visual methods. These plasmonic sensing methods, centered around GNPs, have shown considerable promise in detecting glioma-specific and glioma stem cell markers [56,57]. They offer high sensitivity and specificity, presenting a non-invasive means of diagnosing gliomas through the analysis of blood or cerebrospinal fluid samples [57,58]. While GNPs and magnetic particles (MPs) are the primary nanomaterials employed in colorimetric tumor marker detection methods, novel nanomaterials are continually being developed and integrated into these assays. This expansion of nanomaterial options promises to further enhance the capabilities of colorimetric immunoassays for tumor marker detection, simplifying the assay methodology and pushing the limits of detection sensitivity to even greater levels [59].

In a recent study by Choate and colleagues in 2023, a promising technique for the rapid and extraction-free detection of the R132H isocitrate dehydrogenase 1 (IDH1) mutation in glioma samples was introduced. This mutation serves as a prognostic biomarker and is particularly relevant for glioma prognosis when combined with aggressive surgical resection. The study established the feasibility of a method called colorimetric peptide nucleic acid loop-mediated isothermal amplification (CPNA-LAMP) for this purpose. CPNA-LAMP relies on four conventional LAMP primers, a blocking PNA probe specific to the wild-type sequence, and a self-annealing loop primer targeting the single-nucleotide variant. This approach selectively amplifies the DNA sequence containing the IDH1-R132H mutation. The assay’s effectiveness was validated using synthetic DNA samples with IDH1-WT or IDH1-R132H mutations, as well as cell lysates from U87MG cells with wild-type or IDH1-R132H mutations. Additionally, tumor lysates from archived patient samples with a known IDH1 status, determined using immunohistochemistry (IHC), were analyzed. Notably, the CPNA-LAMP technique demonstrated its capability to swiftly detect the R132H single-nucleotide variant in tumor samples, all within a time frame of under 1 h and without the need for nucleic acid extraction. The visual interpretation of the results relied on a pink-to-yellow color change, providing a simple and accessible means of detection. Further validation through agarose gel electrophoresis confirmed the accuracy of the method. The results of the study indicated a 100% concordance with the IHC results, even in cases where the single-nucleotide variant was localized to specific portions of the tumor. Importantly, the CPNA-LAMP technique exhibited a high specificity, with no instances of false positives or false negatives during the testing of the tumor lysates [60].

### 2.2. Electrochemical Biosensors for Brain Cancer Diagnosis Using Tumor Biomarkers

Electrochemical biosensors stand as versatile tools capable of detecting electrochemical reactions and precisely measuring the changes that occur at the electrode surface. This capability relies on modulating the number of transported ions on the electrode surface, establishing a direct relationship between analyte concentration and the resulting electrochemical signal [61,62,63]. By leveraging electronic transmissions, these biosensors play a pivotal role in measuring and detecting various biological molecules. In electrochemical biosensor systems, three integrated components are crucial for the effective design of: (i) a recognition element for interacting with the analyte; (ii) a signal transducer to generate measurable signals from the analyte–biomolecular layer interaction; and (iii) an electronic system for data management. The sensitivity and specificity of the sensing molecules, particularly the widely employed antibody molecules, enzymes, and synthetic molecular recognition elements, such as short DNA fragments, play a pivotal role in the success of biosensor devices. Depending on the biorecognition molecules used, biosensors can be categorized as immunobiosensors, enzymatic biosensors, and genobiosensors (nucleic acid biosensors) [64]. To date, a multitude of promising electrochemical strategies have been applied for the detection of cancer biomarkers [65]. These encompass various voltammetric techniques, including cyclic voltammetry, linear sweep voltammetry, differential pulse voltammetry, square wave voltammetry, and stripping voltammetry. Additionally, amperometry and impedimetry have been utilized in this context.

Voltammetric biosensors for cancer biomarker detection necessitate two- or three-electrode electrochemical cell systems coupled with a potentiostat, enabling the application of a potential and the subsequent measurement of the obtained current. The careful design of the biosensor’s surface structure is imperative for analyte recognition, ensuring specific interactions while suppressing non-specific ones. The detection limit, ranging from femtomolar (fM) to nanomolar (nM), is contingent on biosensor components, such as gold nanoparticles, carbon nanotubes, magnetic particles, and quantum dots. The integration of nanomaterials in biosensor construction capitalizes on their unique electronic, optical, mechanical, and thermal properties. Gold nanoparticles (GNPs), GNPs nanocomposites, carbon nanomaterials (graphene, carbon nanotubes, and nanowires), redox molecules, dendrimers, quantum dots, sol-gels, polymer matrices, and techniques like self-assembled monolayers and layer-by-layer play pivotal roles in enhancing sensitivity and surface stability [66].

Amperometric biosensors operate through a sequential process: (i) an antibody is labeled with an electro-active species, such as an enzyme or nanoparticles; (ii) the binding of this structure with the analyte through an intermediate primary antibody; and (iii) the quantification of the analyte concentration by applying a potential and measuring the resultant current. The efficacy of amperometric biosensors is closely tied to the electrode properties, given that the signal response occurs proximally to the sensor’s electrode surface [67].

Electrochemical impedance spectroscopy (EIS) has emerged as a vital electrochemical surface characterization technique for analyzing electrode kinetics and electrode–analyte binding characteristics. EIS measurements entail observing the current response to the application of an AC voltage on a constant DC bias for signaling processes [68]. Immobilizing biomaterials, including enzymes, antigens/antibodies, or DNA sequences, onto electrode surfaces induces alterations in capacitance and interfacial electron transfer resistance.

Recent studies have highlighted the potential of electrochemical biosensors for the early diagnosis of GBMs. The application of electrochemical biosensors in GBM diagnosis holds immense promise for revolutionizing early detection strategies. As technology continues to advance, these biosensors are likely to play a pivotal role in improving the precision, speed, and accessibility of glioblastoma diagnostics, ultimately contributing to enhanced patient care and outcomes.

In a 2020 study led by Sun and colleagues, an electrochemical biosensor was developed utilizing Zr-based metal–organic frameworks (Zr-MOFs) for the detection of glioblastoma-derived exosomes with practical applications. Glioblastomas (GBMs), one of the most fatal brain tumors, pose challenges in early diagnosis due to complex oncogenic alterations and the blood–brain barrier (BBB). GBM-derived exosomes, containing specific markers, can traverse the BBB, serving as potential non-invasive biomarkers for early GBM diagnosis. The proposed electrochemical biosensor, sensitive and label-free, incorporates a peptide ligand capable of specifically binding to the human epidermal growth factor receptor (EGFR) and EGFR variant III mutation (EGFRvIII), both overexpressed on GBM-derived exosomes. Simultaneously, Zr-MOFs, encapsulated with methylene blue, can adhere to the exosome surfaces due to the interaction between Zr4+ and the intrinsic phosphate groups outside of the exosomes. The exosome concentration is directly quantified by monitoring the electroactive molecules inside the MOFs, ranging from 9.5 × 10^3^ to 1.9 × 10^7^ particles/μL, with a detection limit of 7.83 × 10^3^ particles/μL. The proposed biosensor has the ability to differentiate GBM patients from healthy groups, showcasing its significant potential for early clinical diagnosis [69].

In a study conducted by Lin et al. in 2021, a highly sensitive and rapid analytical technique was introduced for profiling circulating exosomes directly from the serum plasma of patients with glioblastomas. The methodology involved labeling exosomes with target-specific metal nanoparticles and detecting them using a miniaturized integrated magneto-electrochemical sensing system. Notably, this integrated system exhibited superior detection sensitivity compared to the current methods, allowing for the differentiation of GBM exosomes from exosomes derived from non-tumor host cells. The study also demonstrated that circulating GBM exosomes could be utilized for analyzing primary tumor mutations and serve as a predictive metric for treatment-induced changes. The platform proposed in the study has the potential to offer an early indicator of drug efficacy and function as a molecular stratifier in human clinical trials. This innovative approach provides valuable insights into the improvement of the monitoring of therapeutic responses in GBM patients [70].

### 2.3. Optical Coherence Tomography

Optical coherence tomography (OCT) has emerged as a promising technology for the in vivo, high-resolution, and real-time imaging of the brain tissue during tumor surgery [71]. Given the absence of a proper intraoperative visualization method, there is a growing interest in exploring alternative technologies, and OCT stands out as a compelling option. OCT is based on backscattering, offering a resolution of 0.004 mm^3^ and a penetration depth of less than 2 mm, making it suitable for scanning volumes of 8–16 mm^3^ [72]. It also supports label-free imaging. OCT has several potential applications in neurosurgery: In intraoperative brain imaging, OCT can provide real-time feedback to surgeons during brain tumor surgery. It helps in delineating the boundaries of infiltrative brain tumors within the surrounding tissues and assessing the extent of the damage to the white matter [72,73,74]. In histopathological studies, OCT can be used for rapid tissue type determination in fresh specimens. It aids in differentiating between tumorous and non-tumorous tissues [72,75]. For stereotactic procedures, OCT is valuable for guiding biopsies accurately [72].

OCT offers numerous advantages compared to other intraoperative technologies, including high resolution, rapid imaging, cost-effectiveness, the absence of the need for contrast agents, non-invasiveness, and ease of use. It can be integrated into surgical microscopes or endoscopes [76]. Moreover, OCT can provide considerable functional information about tissues using functional OCT modalities, including Doppler OCT (DOCT), OCT angiography (OCTA), spectroscopic OCT (SOCT), and molecular imaging OCT [76]. These functional extensions offer insights into tissue function and vascular structures. For the advanced visualization of structureless tissues, such as the brain tissue, polarization-sensitive (PS) or cross-polarization (CP) OCT methods are particularly promising. These methods can detect polarization state changes in tissues, generating tissue-specific contrast and improving the visualization of structures like myelinated nerve fibers [77]. Doppler OCT (DOCT) and OCT angiography (OCTA) are functional imaging techniques used to quantify the speed of moving particles within tissues, enabling the acquisition of high-resolution structural images of the vascular network. OCTA, in particular, holds significant promise for clinical use by neurosurgeons, although its precise advantages continue to undergo comprehensive exploration. It provides a range of distinctive attributes that prove invaluable. Notably, it allows for the visualization of the intricate cerebral microvasculature and offers exceptional spatial resolution, depth-resolved data, and the ability to quantify actual blood flow rates, all while being entirely non-invasive [78].

The ability of OCT to distinguish between tumorous and non-tumorous tissues is crucial for achieving high-quality tumor resection while preserving essential white matter tracts. Several studies have assessed the sensitivity and specificity of OCT in this regard, with varying results depending on the glioma grade and the assessment methods used. Several studies have investigated the sensitivity and specificity of optical coherence tomography (OCT) in distinguishing between tumorous and non-tumorous brain tissues [72].

In a study conducted in 2022 by Paul Strenge and his colleagues, the challenging task of defining tumor borders in cases of glioblastoma multiforme during surgical resection was addressed. The primary goal of such resections is to completely remove the tumor while preserving healthy brain tissue. Optical coherence tomography (OCT) has gained prominence as a tool to distinguish between the white matter and tumor-infiltrated white matter. Building on this progress, the researchers created a dataset that included corresponding ex vivo OCT images acquired using two OCT systems with distinct properties, including differences in wavelength and resolution. Each OCT image was meticulously annotated with semantic labels, distinguishing between the white matter, the gray matter, and three different stages of tumor infiltration [79].

The dataset not only facilitated a comparison of each system’s ability to identify the various tissue types encountered during tumor resection but also enabled a multimodal tissue analysis, simultaneously evaluating the OCT images from both systems. To enhance the accuracy of tissue classification, a convolutional neural network with a Dirichlet prior was trained, allowing for the capture of prediction uncertainty. The introduction of this innovative approach significantly improved the sensitivity of tumor infiltration identification, increasing it from 58% to 78% for data with low prediction uncertainty compared to a previous single-modal approach. This work demonstrated the potential of multimodal OCT and advanced machine learning techniques to refine the assessment of glioblastoma tissue during surgery, promising improved outcomes for patients [79].

Additionally, in a study conducted by Han and Cha (2020), a novel technique for intraoperative imaging during brain tumor surgery was introduced. This technique involves near-infrared time-domain reflectometric common-path optical coherence tomography using a bare-fiber probe directly mounted on a scanning galvanometer. The key innovation in this approach is the common-path setup, which offers several advantages, including the flexibility to adjust the optical path length as needed, the use of a disposable fiber probe, and the elimination of a dedicated reference optical path. These improvements simplify the imaging process and enhance its practicality for surgical applications. The experimental results from this study revealed the remarkable capability of the proposed method to effectively discriminate between the brain tumor tissue and normal tissue in mouse brains. Importantly, this discrimination was achieved in real time, and the imaging covered a wide area, providing valuable insights during surgery [80].

In a prospective study involving 18 patients, researchers explored the utility of full-field optical coherence tomography (FF-OCT) for brain tumor diagnosis. The study focused on various brain pathologies, including temporal chronic epileptic parenchyma, and brain tumors, such as meningiomas, low-grade and high-grade gliomas, and choroid plexus papilloma. The FF-OCT method successfully identified a subpopulation of neurons, myelin fibers, and central nervous system (CNS) vasculature. It could distinguish between the cortex and white matter, but it could not visualize individual glial cells, such as astrocytes (normal or reactive) and oligodendrocytes. Notably, this study demonstrated promise in assessing the margins of tumorous glial tissue and epileptic regions, offering a potential advancement in neurosurgical diagnostics [81].

The study by Yashin et al. (2019) conducted both ex vivo and in vivo assessments on 30 glioma patients (grades 2–4) and 17 glioma patients (grades 2–4), respectively. Their study included the examination of the cortex, white matter, and tumor tissues. Through a qualitative assessment, they reported sensitivity and specificity values ranging from 82% to 85% and from 92% to 94% for low-grade gliomas (LGGs) and high-grade gliomas (HGGs), respectively [82]. Furthermore, Kut et al. (2015) conducted a study both in vivo using mice and ex vivo on human glioma patients (grades 2–4). Their research focused on the cortex, white matter, and tumor tissues, utilizing quantitative color-coded maps. They reported a sensitivity and specificity of 100% and 80% for low-grade gliomas (LGGs) and 92% and 100% for high-grade gliomas (HGGs), respectively [83]. Bohringer et al. (2009) conducted an in vivo study involving nine patients with gliomas of grades 2 to 4. Their research examined the cortex, white matter, and tumor tissues using qualitative and quantitative assessments. While specific sensitivity and specificity data were not provided, their study correlated the optical tissue analysis score with histological results, demonstrating a strong relationship (χ^2^ test; r = 0.99) [64]. In summary, these studies collectively demonstrate the potential of OCT in distinguishing between tumorous and non-tumorous brain tissues, with varying sensitivity and specificity values depending on the grade of the gliomas and the assessment techniques employed.

Several studies have explored the utility of optical coherence tomography (OCT) in differentiating white matter from tumorous tissue in the brain. To facilitate this differentiation, visual assessment criteria based on two-dimensional OCT images have been proposed [82,84]. These criteria have been developed by various research groups, aiming to define accurate parameters for distinguishing between tumorous and normal brain tissues based on OCT signal intensity characteristics.

Further refinements in the criteria for distinguishing tumorous tissue from white matter were made with the use of cross-polarization OCT (CP OCT) devices. While OCT images from tumors may exhibit variability, common features were identified, contributing to a high level of interrater agreement. The main criterion in these refinements became the intensity of the OCT signal in both co- and cross-polarizations. Additionally, signal homogeneity or heterogeneity and the uniformity of signal attenuation along the lower boundary in the co-polarization images were considered as supplementary factors. Furthermore, the quantitative evaluation of OCT data, including the calculation of the attenuation coefficient, demonstrated a higher diagnostic accuracy in distinguishing tumors from white matter compared to visual assessment. Optical maps reflecting the distribution of attenuation coefficient values throughout the image provided enhanced contrast results, facilitating an improved delineation between the normal white matter and tumor tissue [85].

Notably, the myelin content greatly influences OCT signal attenuation in the white matter, allowing for the assessment of myelinated fiber density and arrangement. Preliminary investigations have shown that damage to myelinated fibers due to tumor invasion results in changes in OCT signal attenuation, allowing for both qualitative and quantitative assessment. This involves observing a slowdown in signal attenuation in both polarizations, leading to a decrease in the calculated attenuation coefficient values. Consequently, optical maps display distinct differences reflecting the state of the white matter, which can be crucial for determining the extent of myelinated fiber damage [82].

In a recent study conducted in 2023, the focus was on improving the precision of brain tumor resections by effectively distinguishing the regions with damaged myelinated fibers from the tumor tissue and the normal white matter. The study highlighted the success of employing cross-polarization (CP) optical coherence tomography (OCT) for this purpose. The research involved 215 brain tissue samples collected from 57 patients with brain tumors. The study’s results demonstrated that the visual inspection of structural CP OCT images effectively discerned areas within the white matter with damaged myelinated fibers, enabling a clear differentiation from the normal white matter and the tumor tissue. The attenuation coefficients proved valuable in distinguishing various brain tissue types, with significantly lower values detected in areas with damaged myelinated fibers compared to the normal white matter. However, the application of color-coded optical maps emerged as a more promising approach, as it combined the objectivity of optical coefficients with the clarity of visual assessment. This approach substantially improved diagnostic accuracy compared to the visual analysis of structural OCT images [86]. Bohringer et al. conducted a qualitative analysis of OCT images from glial tumors with varying malignancies, identifying signal homogeneity as a key differential criterion. They found that the tumor tissue and the surrounding peritumoral region (infiltration zone) exhibited heterogeneous signals, in contrast to the homogeneous signals that are typical of the normal brain tissue. Importantly, their study demonstrated a strong correlation between OCT signal characteristics and histological findings, reinforcing the diagnostic potential of OCT [84].

In differentiating between the tumorous tissue and the gray matter, challenges arise from closely approximated OCT signal parameters. At present, limited research has been dedicated to detecting tumor infiltration within the gray matter and basal ganglia. Gray matter regions, such as the putamen, globus pallidum, thalamus, and subthalamic nuclei, are particularly challenging due to their deep-seated location in the brain. Detecting differentiation within the hippocampus, which exhibits a lower scattering strength than the cortex, is expected to be difficult. Still, some studies have demonstrated the differences between the white and gray matters, enabling OCT to be applied effectively in specific contexts [87].

In a study conducted by Strenge and colleagues in 2022, the focus was on demarcating the boundary between the brain tissue and tumor tissue through the application of OCT in conjunction with neural networks trained on prior data. Recent advances have demonstrated that the discrimination between white matter and tumor-infiltrated white matter, based on OCT data, can be achieved with a high degree of accuracy. However, the presence of gray matter in the context of tumor resection poses a significant challenge, as it exhibits optical properties similar to those of tumor infiltration. This similarity complicates the task of classifying tumor tissue using optical coherence tomography. To address this challenge, a semantic segmentation approach was employed, utilizing a convolutional neural network to distinguish healthy brain tissue from tumor-infiltrated brain tissue. A dataset was meticulously curated, comprising ex vivo OCT B-scans obtained from a swept-source OCT system with a central wavelength of 1300 nm. Each OCT B-scan was indirectly annotated by transferring histological labels from a corresponding H&E (hematoxylin and eosin) section onto it. These labels provided differentiation between the white matter, gray matter, and tumor infiltration. A noteworthy feature of the network’s output was its modeling to a Dirichlet prior distribution, allowing for the incorporation of prediction uncertainty. This novel approach yielded impressive results, achieving an intersection over union (IoU) score of 0.72 for healthy brain tissue and 0.69 for highly tumor-infiltrated brain tissue when considering only confident predictions. In summary, the study by Strenge and colleagues showcased a cutting-edge methodology that leveraged OCT data and neural networks to effectively differentiate between healthy brain tissue and areas infiltrated by tumors, even in the presence of challenging gray matter, demonstrating the potential for improved precision in neurosurgical procedures [79].

Furthermore, by harnessing the power of optical coherence angiography, researchers can delve into the intricate vascular characteristics of brain tumors. This enables a deeper understanding of how blood vessels function within and around these tumors, providing crucial insights that can influence the development of more effective diagnostic and therapeutic strategies. In this context, optical coherence angiography serves as a valuable tool for investigating the complexities of brain tumor vasculature, ultimately advancing our knowledge in the fight against these challenging medical conditions. The study conducted by Farah Andleeb and colleagues in 2021 aimed to address challenges in monitoring therapeutic efficacy for malignant gliomas and characterizing tumor vasculature. They utilized optical coherence angiography to examine vasculature features within and around brain tumors in a murine xenograft brain tumor model. The analysis included factors such as fractional blood volume, vessel tortuosity, diameter, orientation, and directionality. The study involved imaging five murine tumor models with human glioblastoma cells injected into their brains. After allowing the tumors to grow for four weeks, they were imaged using optical coherence tomography. The results revealed significant differences in vascular characteristics. The blood vessels outside the tumor exhibited a higher fractional blood volume compared to those within the tumor. The vessels within the tumor were found to be more tortuous or twisted compared to those outside the tumor. Additionally, the vessels near the tumor’s edge displayed a tendency to direct inward toward the tumor, while normal vessels had a more random orientation. In conclusion, the quantification of vascular microenvironments within brain gliomas provides valuable functional vascular parameters that can contribute to diagnostic and therapeutic research [87].

Stereotactic biopsies are a common neurosurgical procedure for diagnosing glial brain tumors and intracranial lymphomas [88]. Nevertheless, there is a risk of acquiring non-diagnostic samples, thus necessitating repeated surgeries or intraoperative neuropathological assessments to improve diagnostic accuracy. These techniques have their limitations, such as a high time consumption and increased risks [88,89].

The recently proposed stereotactic OCT probes offer an innovative solution for enhancing the accuracy and safety of biopsies. These probes are designed to provide real-time optical biopsies and to detect blood vessels in the biopsy area, thus minimizing the need for intraoperative histopathological examination and reducing the risk of intracerebral hemorrhages. The OCT probe can be integrated into a standard biopsy needle, allowing for the precise monitoring of needle placement and immediate tissue analysis in the biopsy area. Such advancements in stereotactic OCT probes have the potential to make procedures safer, more accurate, and less invasive [90].

The 2019 study conducted by Kiseleva and colleagues aimed to advance minimally invasive techniques for brain tumor biopsies, utilizing cross-polarization (CP) optical coherence tomography (OCT) to enhance neurosurgical procedures within modern neuro-oncology. The primary objective of their research was to develop a specialized tool by integrating CP OCT technology into a standard biopsy needle, with the goal of improving the precision and safety of stereotactic brain biopsies. The study involved in vivo experiments on healthy rat brains, successfully demonstrating the probe’s capability to detect blood vessels along the brain’s surface as the biopsy needle advanced. Additionally, it showed the probe’s ability to differentiate various tissue types, including cerebral cortex and white matter, as the needle penetrated the brain. While the initial image assessment relied on visual criteria, the study highlighted the potential for heightened sensitivity and specificity in differentiating tissue types and detecting blood vessels through the implementation of CP OCT signal quantification methods. In summary, this research underscored the potential of CP OCT as an effective tool for guiding OCT-assisted stereotactic brain tumor biopsies, offering the promise of improving the precision and safety of these neurosurgical procedures [90].

Furthermore, a study conducted by Ramakonar and his colleagues (2018) addressed the significant issue of intracranial hemorrhage that can occur during brain needle biopsies, posing a potential risk to the nearby blood vessels. There is a lack of intraoperative technology available to reliably identify blood vessels at risk of damage during these procedures. To tackle this problem, the researchers developed an “imaging needle” equipped with a miniaturized optical coherence tomography (OCT) probe. This imaging needle allows for the real-time visualization of nearby blood vessels while differentiating between blood flow and the surrounding tissue. In a clinical study involving 11 patients, the imaging needle demonstrated the ability to intraoperatively detect blood vessels with a diameter exceeding 500 μm. The results showed a high sensitivity of 91.2% and an impressive specificity of 97.7%. The findings suggested that imaging needles have the potential to be a valuable tool in various neurosurgical needle interventions, providing surgeons with a means to identify and avoid damaging blood vessels during brain biopsies, thus enhancing patient safety and minimizing the risk of intracranial hemorrhage [91].

In the future, glioma research will continue to benefit from the capabilities of OCT. Researchers are expected to further refine and expand OCT methods, exploring new modalities and applications. The incorporation of artificial intelligence and machine learning is likely to enhance tissue differentiation, thus increasing the precision of tumor resections. In summary, OCT not only offers immediate benefits in glioma surgery but also opens the door to a promising future where neurosurgeons can achieve even higher levels of precision and improved patient outcomes. The evolution of OCT in glioma research underscores its significance as a valuable tool in the ongoing battle against these challenging medical conditions.

### 2.4. Surface-Enhanced Raman Spectroscopy

Spectroscopic techniques based on Raman scattering have been proven to be powerful diagnostic tools providing objective biochemical fingerprints to distinguish among the normal, benign, and cancer tissues of many organs. The core principle of Raman spectroscopy involves the interaction of light with matter, resulting in photons being scattered either elastically (Rayleigh scattering) or inelastically through the Raman effect.

Raman spectroscopy (RS) offers several key advantages over the traditional diagnostic methods, making it a valuable tool in various applications. RS provides objective biochemical information about the constituents of normal, benign, and cancer cells, and it has the capability to identify different cancer markers in a single measurement. One of the prominent advantages of RS over other spectroscopic techniques, such as infrared (IR) spectroscopy, is its ability to operate without interference from water. This is especially critical for applications involving live-cell analysis, human tissue examination, and in vivo studies. Additionally, RS boasts a high spatial resolution, which enhances its versatility and utility [92,93,94].

However, there is a notable limitation associated with Raman-based optical methods—the inherently weak Raman signal. When light interacts with matter, it undergoes either elastic scattering (Rayleigh scattering) or inelastic scattering through the Raman effect. Raman scattering involves a relatively small fraction of photons, approximately one in ten million, exchanging energy with molecules via vibrational transitions. This process leads to the production of Stokes and anti-Stokes Raman scattering. Stokes Raman scattering occurs when the scattered photons possess lower frequencies than the incident ones, causing the molecule to shift to a higher vibrational state. In contrast, anti-Stokes Raman scattering results from scattered photons with higher frequencies, leading to a lower vibrational state. The specific vibrational transitions are unique to the molecular composition, resulting in a distinctive Raman spectrum that offers intricate structural and chemical information [94,95].

Despite the intrinsic inefficiency of Raman scattering, several methods have been developed to enhance its signal. Resonance Raman scattering involves using an excitation laser wavelength that aligns with the electronic transitions of the molecules, amplifying the Raman signal by factors ranging from 10^2^ to 10^6^. For even more substantial enhancement, positioning molecules near plasmonic materials, such as metal nanostructures, leverages both the interactions of light with the molecules and light–metal interactions. This synergy significantly bolsters the inelastic scattering efficiency, giving rise to surface-enhanced Raman spectroscopy (SERS), a phenomenon known for its remarkable signal amplification and utility in various applications [96].

SERS relies on the excitation of surface plasmons on a rough metallic surface, which results in highly amplified Raman signals. Metals, like Ag, Au, and Cu, are commonly used in SERS, with nanoparticles (NPs) of these metals exhibiting intense absorption bands in the UV-Vis region. The excitation wavelength in SERS experiments must be in resonance with the absorption properties of the synthesized NPs to achieve the strongest enhancement. This unique feature makes SERS a powerful multiplexing technique with high sensitivity, even in the picomolar range [96,97,98].

Gliomas are known for their high heterogeneity, and they exhibit various molecular markers that can serve as diagnostic, predictive, and prognostic indicators for these tumors. Among these markers, isocitrate dehydrogenases (IDHs) play a central role in glioma characterization and have become a routine part of histopathological diagnosis, drug sensitivity assessment, and prognosis evaluation as outlined in the WHO CNS5 2021 guidelines [39]. It is important to note that, while only 12% of glioma patients carry the IDH mutation, discovered after genomic analysis, those with the IDH mutation tend to have a more favorable survival prognosis. Raman analysis reveals distinctive differences in the spectral features of scattered light from various cellular components, including lipids, collagen, DNA, cholesterol, and phospholipids [99,100]. Notably, a study involving 38 unprocessed samples, comprising a total of 2073 Raman spectra, demonstrated the potential of RS to differentiate between IDH-mutant (IDH-MUT) and IDH-wildtype (IDH-WT) gliomas with an accuracy and precision of 87%. Machine learning techniques, such as the radial basis function support vector machine (RBF-SVM), play a pivotal role in achieving this differentiation [101].

Recent innovations extend to the intraoperative detection of blood vessels during neurosurgery, enhancing surgical precision and patient care. SERS (surface-enhanced Raman spectroscopy) and optoacoustic tomography are promising in guiding brain tumor resection, with dual-modal approaches earning recognition for their potential clinical translation [102].

Notably, SERS facilitates the targeting of glioblastoma tissues using a hand-held Raman scanner in genetically engineered mouse models, while stimulated Raman histology, coupled with convolutional neural networks, achieves a remarkable 100% classification accuracy and expedites brain tumor diagnosis within the operating room. This advancement outpaces conventional techniques, significantly reducing diagnosis time and thus improving patient care. A study was conducted to assess the effectiveness of a hand-held Raman scanner guided by surface-enhanced Raman scattering (SERS) nanoparticles in identifying the microscopic tumor extent in a genetically engineered RCAS/tv-a glioblastoma mouse model. In a simulated intraoperative setting, both a static Raman imaging device and a portable hand-held Raman scanner were tested. The results demonstrated that the SERS image-guided resection was more accurate than the resection relying solely on white light visualization. Both methods complemented each other, and comparison with histological analysis confirmed that SERS nanoparticles precisely outlined the tumor boundaries [103].

A study conducted by Burgio and colleagues in 2020 explored the application of SERS to improve the visualization of GBM tumor borders during surgery. Their objective was to overcome the challenge of gold nanoparticles (GNPs) aggregating or binding non-specifically to cells, which compromises the precise discrimination between tumor and healthy cells. To address this issue, the researchers focused on optimizing the surface chemistry of GNPs by balancing inert and active targeting functionalities. The study involved GNPs with varying ratios of Raman reporters, polyethylene glycol (PEG), and antibodies targeting the epidermal growth factor receptor, which is overexpressed in GBM cells. The researchers meticulously examined how these ratios influenced GNP performance, taking into account factors such as colloidal stability, sensitivity, and non-specific binding. They determined that the optimal GNP functionalization involved 50% Raman reporter surface coverage and 3% antibody surface coverage. This particular configuration prevented GNP aggregation, reduced non-specific binding, and provided sufficient Raman sensitivity for the rapid and clear differentiation between GBM tumor and non-tumoral cell lines in vitro. Furthermore, the researchers discovered that an excess of antibodies did not improve GNP binding to tumor cells; instead, it reduced the conjugation efficiency by 35%. These findings offer a stable and non-quenching alternative for GBM visualization, surpassing the current state-of-the-art technique of fluorescence-guided surgery [104].

Detecting brain-cancer-specific biomarkers in the blood is a challenging task, primarily due to the limited exchange of biomolecules between the bloodstream and the brain. In a more recent study, Premachandran and colleagues introduced a novel SERS platform based on Ni-NiO designed to detect molecules present in the blood, thus allowing for the accurate identification of primary and secondary tumors. The hybrid SERS substrate created in this study combined the electromagnetic enhancement from metallic Ni with chemical enhancement via a charge transfer mechanism. This innovative method relied on Raman molecular profiles obtained from a minimal working volume of 5 µL of sera. In the Raman spectrum of brain cancer, distinctive peaks associated with lipids, fatty acids, and proteins were identified. To validate the specificity of this platform for cancer detection, the molecular signatures of brain cancer sera were compared to those of breast, lung, and colorectal cancers. Furthermore, this method was capable of pinpointing the exact tumor location based on the presence of specific species, such as glycogen, phosphatidylinositol, nucleic acids, and lipids [105].

Another promising approach was presented by Kircher and colleagues, who combined SERS, photoacoustic imaging (PA), and magnetic resonance imaging (MRI) to achieve the highly precise visualization of brain tumor margins. This was achieved through the use of gold nanotags functionalized with Gd organometallic complexes [106]. This innovative approach, which integrates endoscopic, photoacoustic, and Raman imaging capabilities, opens the door to the potential clinical translation of the MPR approach (magnetic resonance imaging–photoacoustic imaging–Raman imaging nanoparticle).

In a separate study conducted by Neuschmelting and colleagues in 2018, the potential of SERS and optoacoustic tomography for intraoperative brain tumor delineation was addressed, aiming to enhance surgical care. The study aimed to overcome the persistent challenge of visualizing glioma margins during intraoperative procedures, a crucial factor in achieving complete tumor resection and improving the clinical outcomes of glioblastoma (GBM) patients. The research involved the development of a strategy that included a newly designed gold nanostar synthesis method, Raman reporter chemistry, and a silication technique to create dual-modality contrast agents for simultaneous surface-enhanced resonance Raman scattering (SERRS) and multispectral optoacoustic tomography (MSOT) imaging. In the experimental phase, brain-tumor-bearing mice were intravenously injected with the SERRS–MSOT–nanostars agent, and sequential in vivo MSOT imaging was conducted, followed by Raman imaging. MSOT successfully provided the accurate three-dimensional visualization of GBMs with a high level of specificity. The MSOT signal correlated effectively with the SERRS images. Importantly, SERRS, known for its uniquely sensitive and high-resolution surface detection capabilities, served as an ideal complementary imaging modality to MSOT, which excels in real-time deep-tissue 3D imaging. The dual-modality SERRS–MSOT–nanostar contrast agent described in this study demonstrated its potential to precisely delineate the extent of infiltrating GBMs through Raman and MSOT imaging in a clinically relevant murine GBM model. This approach is promising for advancing image-guided brain tumor resection, potentially leading to improved outcomes for patients [107].

Distinguishing between different types and grades of gliomas is vital for optimizing patient care, treatment planning, prognosis assessment, and advancing medical research. It enables healthcare providers to make informed decisions that ultimately lead to improved outcomes and quality of life for individuals affected by gliomas. The research conducted by Jingwen Li (2020) focused on the label-free discrimination of glioma brain tumors at different stages using surface-enhanced Raman scattering (SERS). The study utilized substrates consisting of silver nanoparticles decorated on silver nanorods, known as AgNPs@AgNRs. These AgNPs@AgNR substrates demonstrated remarkable SERS performance, boasting an impressive enhancement factor of up to 1.37 × 10^9^, surpassing the capabilities of other SERS-active silver nanoparticle and silver nanorod substrates. Through the integration of AgNPs@AgNR substrates with principal component analysis (PCA), the research team achieved a rapid differentiation between healthy brain tissue and gliomas at various stages. The spectra obtained from the tissue samples revealed pronounced spectral differences, enabling the distinction between healthy regions and areas affected by gliomas. One of the most notable distinctions in the SERS spectra was the reduction in the ratio of two characteristic peaks at 653 and 724 cm^−1^, when comparing healthy brain tissue to gliomas at different stages. Additionally, the utilization of three-dimensional PCA allowed for a clear differentiation between healthy brain tissue and grade II gliomas (considered as low-grade) as well as between grade III and grade IV gliomas (considered as high-grade). The preliminary results indicated that the SERS spectra based on AgNPs@AgNR substrates show great potential for a rapid and straightforward identification, thanks to the uncomplicated specimen preparation and high-speed spectral acquisition involved in the process. This innovative method is promising for making significant contributions to the field of glioma diagnosis and characterization [108].

In a further investigation conducted by Bury and colleagues, a comprehensive analysis was conducted on 29 brain tissue samples acquired during surgical procedures. The study introduced an innovative approach, employing a handheld Raman probe in conjunction with gold nanoparticles, to detect primary and metastatic brain tumors in fresh brain tissue sent for intraoperative smear diagnosis. The fresh brain tissue samples designated for this purpose underwent testing using the handheld Raman probe after the application of gold nanoparticles. The Raman spectra obtained were used to develop predictive models for sensitivity and specificity in diagnostic outcomes. The results demonstrated the capability to distinguish between primary and metastatic tumors, particularly in the case of normal and low-grade lesions. The study achieved impressive levels of accuracy, sensitivity, and specificity for different tumor types based on smear samples [109].

In another notable contribution to the field, Hollon and colleagues demonstrated the effectiveness of stimulated Raman histology as a powerful technique for near real-time intraoperative brain tumor diagnosis. By integrating a convolutional neural network (CNN) with stimulated Raman histology, researchers achieved an impressive 100% classification accuracy. Leveraging recent advancements in deep learning, the CNN was trained on a dataset of over 2.5 million stimulated Raman histology images, enabling a rapid brain tumor diagnosis in the operating room in under 150 s—considerably faster than the traditional techniques. This clinical trial’s outcome underscores the potential of stimulated Raman histology as a complementary approach to tissue diagnosis, which can significantly enhance the care provided to brain tumor patients [110].

The study by Desroches et al. addresses the limitations of current cancer diagnosis methods, which rely on blind needle biopsies that can lead to targeting errors and inaccurate sampling due to the heterogeneity of tumors. These issues often result in non-diagnostic or poor-quality samples, elevating patient risks and necessitating repeated biopsies. To improve the accuracy of cancer targeting and reduce patient risks, the researchers developed an in situ intraoperative cancer detection system based on high-wavenumber Raman spectroscopy. This optical device was seamlessly integrated into a commercially available biopsy system, allowing for the analysis of the tumor tissue’s molecular properties before the actual tissue-harvesting procedure, without disrupting the surgical workflow. Through a dual validation approach, the study demonstrated that high-wavenumber Raman spectroscopy can effectively detect dense cancer tissue with over 60% cancer cells in situ during surgery. The system showed a sensitivity of 80% and a specificity of 90%. Additionally, the research extended the application of this system to a swine brain biopsy model. These findings pave the way for the clinical implementation of this optical molecular imaging method, promising high-yield and safe, targeted biopsies. This technology has the potential to significantly improve the accuracy of cancer diagnosis and reduce the risks associated with blind biopsy procedures [111].

Collectively, the aforementioned findings pave the way for the translation of Raman-based techniques from the research laboratory to clinical applications. This transition is promising, as it can significantly enhance care and outcomes for individuals with brain tumors. The clinical application of Raman-based techniques, exemplified by the use of Raman probes, intraoperative guidance, and machine learning, has the potential to revolutionize neurosurgery and brain tumor diagnosis. These advancements not only offer the potential for improved patient care but also streamline the diagnostic process, rendering it faster and more efficient.

As Raman-based techniques continue to progress and make their way into clinical practice, they have great potential for reshaping the landscape of brain tumor diagnosis and surgical procedures. This transformation stands to benefit both patients and healthcare professionals alike, ushering in a new era of more precise and effective neurosurgical interventions.

### 2.5. Reflectometric Interference Spectroscopy

Reflectometric interference spectroscopy (RIfS) is a sophisticated method that contributes to the detection and assessment of cancer incidence by measuring changes in the refractive index. This technique relies on variations in the amplitude and phase of polarized light, which are influenced by alterations in the refractive index and thickness of an adsorbed layer of the analyte [112].

RIfS is based on the interference of polarized light at the interfaces of transparent thin layers, making it a label-free optical detection method for interactions on surfaces. As light interacts with these thin layers of diverse materials, it is partially reflected and transmitted with minimal absorption. The interference pattern that emerges is a result of the optical thickness, which depends on several factors, including the physical thickness of the layer, its refractive index, the refractive index of the surrounding medium, the incident angle, and the wavelength [112].

The detection principle of RIfS centers on observing the changes in the optical properties of a specific layer system on the top layer. When particles or analyte molecules bind to the sensor surface, it leads to a shift in the interference pattern. This shift results in a time-resolved binding curve, which can be tracked over time to evaluate the binding signal of the analyte molecule on the sensor surface [113]. One remarkable advantage of RIfS is its robustness and simplicity as an optical detection method in chemical and biochemical sensing practices. It provides precise measurements that are less susceptible to temperature changes compared to other methods, like ellipsometry. RIfS is a valuable tool for studying surface interactions, especially in the context of cancer detection, where it can provide real-time insights into binding events at the nanoscale level [112].

RIfS technology has been leveraged to develop real-time applications, including an optical biopsy needle featuring integrated optical fibers at its tip [114]. This innovation opens up exciting possibilities for in vivo applications. Furthermore, researchers have explored the use of RIfS for creating specialized sensors. For example, a sarcosine-imprinted RIfS nanosensor was developed using the spin-coating technique, demonstrating excellent linearity with a correlation coefficient of 0.9622 and a detection limit of 45 nM [115]. Moreover, RIfS has been employed in immuno-sensing applications, as exemplified by a biosensor immobilizing anti-C-reactive protein (CRP) using protein A on an SiN chip [116]. Additionally, innovative solutions based on nanoporous anodic aluminum oxide reflective interferometric sensing have enabled the development of a sensor for volatile sulfur compounds and hydrogen sulfide gas [117].

Research focusing on glioblastoma theragnostics using reflectance interference technology is limited. However, various other reflectance spectroscopy techniques and combined methods have been employed for the investigation of glioblastoma theragnostics. In a recent study conducted by Kerui Li, the research aimed to investigate the viability of using diffuse reflectance spectroscopy (DRS) as a label-free and real-time detection technology for distinguishing between gliomas and noncancerous tissues. To achieve this, the study analyzed 55 fresh specimens from both cancerous and noncancerous brain tissues obtained from 19 different brain surgeries. The data acquired through DRS were subsequently compared with clinically standard histopathology for validation. The research focused on quantitatively obtaining tissue optical properties from the diffuse reflectance spectra and performing comparisons across various types of brain tissues. To facilitate the discrimination between cancerous and noncancerous tissues, the study utilized a machine learning-based classifier. The outcomes of this investigation were quite promising, as the method exhibited a remarkable sensitivity of 93% and a specificity of 95% in distinguishing high-grade gliomas from normal white matter. These results strongly suggest that DRS holds significant potential for label-free and real-time in vivo cancer detection during brain surgery. This innovative approach offers the prospect of improving surgical precision and decision making in the context of brain tumor resections [118]. 

In another study, led by Simon Skyrman, the proof-of-concept research aimed to assess the feasibility of using DRS for the differentiation between glial tumors and healthy brain tissue, particularly in an ex vivo setting. The study involved the acquisition of DRS spectra and histological data from a total of 22 tumor samples and 9 brain tissue samples, obtained from 30 patients. By applying a model derived from the diffusion theory, the research team estimated the content of biological chromophores and scattering features based on the DRS spectra. The results of the study demonstrated significant differences in the DRS parameters between tumor and normal brain tissues. The classification process, employing a random forest algorithm, yielded encouraging outcomes, with sensitivity and specificity values of 82.0% and 82.7%, respectively, for the detection of low-grade gliomas. Additionally, the area under the curve (AUC) presented a noteworthy value of 0.91. These findings underscore the potential utility of DRS, particularly when integrated into a hand-held probe or biopsy needle, for providing an intra-operative tissue analysis. This innovative approach holds promise for enhancing the accuracy of surgical procedures involving glial tumors and facilitating real-time tissue discrimination [119].

In a 2020 study by Baria et al., the researchers explored the in vivo detection of murine glioblastomas utilizing a combination of Raman and reflectance fiber-probe spectroscopies. The study involved a series of steps, beginning with the localization of tumor areas through the detection of EGFP fluorescence emissions. Subsequently, Raman and reflectance spectra were collected from both healthy and tumor tissues. The collected data were subjected to thorough analyses, employing techniques, such as principal component and linear discriminant analyses. These analytical methods aimed to develop a classification algorithm that could effectively distinguish between healthy and tumor tissues. Notably, the results demonstrated a high classification accuracy, with Raman and reflectance spectra achieving accuracies of 92% and 93%, respectively. The combination of these techniques further enhanced the discrimination between healthy and tumor tissues, ultimately achieving an impressive accuracy of up to 97%. In conclusion, these preliminary findings underscore the substantial potential of multimodal fiber-probe spectroscopy for the in vivo label-free detection and delineation of brain tumors. This promising research represents a significant step forward in the journey toward the clinical application and widespread use of fiber-probe spectroscopy in the context of brain tumor diagnosis and surgery [120].

In a study conducted by Hosseinzadeh and colleagues, the research aimed to employ interferometric optical testing for the discrimination between benign and malignant brain tumors. This involved assessing and comparing optical effects to distinguish between the two tissue types. The study analyzed various samples of adult human brain tissues utilizing a Mach–Zehnder interferometer as the optical method, with a subsequent data analysis being performed through the Fourier transform method. The interference patterns generated by the benign and malignant brain tumors were examined to derive the phase distribution characteristic of each tumor. The results revealed notable differences in the phase distribution between benign and malignant brain tissues. Typically, benign samples exhibited phase distributions ranging from 10 to 120 rad, while malignant samples ranged from 10 to 160 rad. Furthermore, the average unwrapped phase distribution measured 63.79 rad for benign tissues and 85.69 rad for malignant tissues. These findings suggest that the proposed laser-based technique, the Mach–Zehnder interferometer method, can serve as a complementary approach alongside histological techniques for distinguishing between benign and malignant brain tumors. It is recommended that the unwrapped phase distribution of tissues be considered as a valuable optical property for the differentiation of various brain tumors [121].

In a 2017 study conducted by Vinh Nguyen Du Le, the researchers explored the potential of a dual-modality optical biopsy for the discrimination of glioblastoma multiforme (GBM) from low-grade gliomas (LGGs) using diffuse reflectance and fluorescence spectroscopy. These non-invasive methods hold promise for enhancing the precision of brain tissue resections during surgery. The study involved the retrieval of optical properties through an experimentally evaluated inverse solution. The key findings included the observation that the scattering coefficient in GBM was, on average, 2.4-times higher than that in LGGs, while the absorption coefficient was 48% higher. Additionally, the ratio of fluorescence to diffuse reflectance at the emission peak of 460 nm was 2.6-times higher for LGGs, whereas reflectance at 650 nm was 2.7-times higher for GBM. One of the noteworthy outcomes of this research was that the combination of diffuse reflectance and fluorescence spectroscopy achieved a remarkable level of sensitivity, reaching 100%, along with a specificity of 90% when distinguishing GBM from LGGs during the ex vivo measurements performed on 22 sites from seven glioma specimens [122].

In prospect, the substantial potential of optical methodologies in glioblastoma research and clinical practice is poised for a transformative impact. As technology advances and research endeavors progress, the imminent integration of these optical techniques within routine clinical protocols becomes foreseeable. The ongoing development of novel applications, specialized sensing devices, and the relentless pursuit of real-time in vivo diagnostics collectively contribute to the expanding repository of knowledge. This increasing knowledge holds the promise of significantly augmenting our capacity to combat glioblastomas. The ongoing journey toward achieving enhanced precision in diagnoses, refined surgical procedures, and improved patient outcomes is in motion, with optical techniques playing a pivotal role in this paradigm shift.

### 2.6. Optical Biosensors

Advancements in optical biosensors that leverage the unique properties of light for detection have ushered in a new era of capabilities, offering real-time monitoring, rapid responses, enhanced accuracy, and heightened sensitivity. These innovative optical biosensors include various types, and their applications have extended to the monitoring and diagnosis of diverse medical conditions, including infectious diseases, cancer, and neurological disorders. Predicated on their design attributes and transducer mechanisms, these optical biosensors span a major range of types, including optic fiber sensors, ring resonators, interferometers, optical waveguides, photonic crystals, fluorescence/luminescence-based sensors, and surface plasmon resonance (SPR) sensors. Importantly, these versatile techniques have found valuable applications in the realm of glioma research [123].

In the context of gliomas, optical biosensors play a pivotal role in advancing the field of theragnostics. Recent studies have been conducted, specifically focusing on their utilization in the context of glioblastomas. These studies represent the most up-to-date research efforts in this critical area of medical science.

Distinguishing between the tumor and peritumoral tissue, particularly in patients with gliomas, holds immense importance in the management of this aggressive brain cancer. This distinction is pivotal for treatment planning, enabling precise surgical strategies that maximize the removal of cancerous cells while minimizing the damage done to essential brain regions. It aids in monitoring treatment response, customizing therapies, and guiding the research and clinical trials to improve glioma outcomes.

Victor Garcia Milan and his research team (2023) conducted a study that unveiled the potential of a plasmonic-based nanostructured biosensor in efficiently distinguishing between tumors and peritumoral tissues, particularly in patients with glioblastomas (GBMs). The experimental procedure was initiated with small tissue samples obtained from GBMs and peritumoral regions being placed onto the biosensor’s surface. Subsequently, the unique imprints left by these tissues were subjected to analysis using an adapted upright microscope, connected to a spectrometer for precise optical measurements. The results were notable, revealing that the biosensing system demonstrated sensitivity and specificity levels of around 80%. Moreover, the area under the curve was calculated to be 0.8779, with a 95% confidence interval spanning from 0.7571 to 0.9988 (*p* < 0.0001). These findings strongly suggest that the biosensor offers a viable and reliable approach for distinguishing a GBM from its peritumoral tissue, marking an essential first step in the road towards the development of an in vivo system capable of performing a real-time differentiation between these tissues, which could significantly aid surgical decision making [124].

However, it is crucial to note that achieving this ambitious goal requires additional efforts. These efforts include optimizing the optical system to enhance specificity, gaining a greater understanding of the biological factors responsible for these optical variations, and conducting comprehensive studies involving larger cohorts of GBM patients undergoing surgical procedures. This promising research paves the way for potential breakthroughs in the field of GBM diagnosis and surgical interventions, holding significant promise for improved patient care and decision support.

Photonic crystals (PhCs) have emerged as an extremely appealing choice for applications in optical data processing (ODP) in recent years, evolving into a pivotal platform for making ODP applications cost-effective. Notably, PhC-based sensors have gained recognition as a promising technique among various photonic-based sensing approaches. Their unique physical attributes, including properties like reflectance and transmittance, have presented outstanding sensitivity levels, leading to precise detection limits, all within the captivating visual spectrum of wavelengths [125].

The following represents a comprehensive comparison of photonic crystal (PhC)-based sensors for the analysis of brain tissue, with a particular emphasis on distinguishing normal brain tissues, various brain tissue subtypes, and detecting tumors and cancers. The presented table presents the data from three distinct studies conducted in the years 2020, 2021, and the most recent study by Mohammed et al. in 2023.

The study conducted by Nouman et al. (2020) involved the theoretical design of a defected 1D PC as a refractive index (RI) sensor intended for detecting brain lesions. Utilizing the transfer matrix method, the research team examined the transmission spectrum of the 1D PC sensor. Specifically, at structural dimensions of Dd = 1.68 μm and θ = 80°, the PC sensor, characterized by the structure [Air/(SiO_2_/PbS)3/D/(SiO_2_/PbS)3/SiO_2_], exhibited high sensitivity. The resultant transmission spectrum of the designed sensor indicated a red shift in the resonant defect peak, altering from 2.063 to 2.497. This shift correlated with the replacement of brain lesions, transitioning from cerebrospinal fluid (CSF) with an RI of 1.3333 to metastasis with an RI of 1.4833. The sensor’s sensitivity ranged from a low value of 2893.333 nm/RIU for metastatic cells to a high value of 3080.808 nm/RIU for oligodendrogliomas.

The figure of merit (FOM) for the sensor reached a notably high value of 6.16 × 10^7^ (1/RIU), signifying its exceptional sensitivity to detect changes in the resonant peak. The designed PC sensor exhibited the capability to detect different brain tissues, such as glioblastomas and multiple sclerosis, characterized by varying densities. An analysis of the transmission spectrum indicated a red shift in the resonant peak as the refractive index changed from 1.4512 to 1.4611 for glioblastomas and from 1.3421 to 1.3641 for multiple sclerosis tissues [126].

The study conducted by Asuvaran and Elatharasan in 2021 introduced 2D PhC sensors with reduced sizes and equipped with improved quality factors when compared to the 1D sensor of 2020. These enhancements strengthened the sensors’ capabilities to effectively distinguish between normal and abnormal brain tissues, all while maintaining consistent detection limits at approximately 10^−3^ RIU. The study demonstrated a robust methodology for identifying various brain lesions, pointing to the evolving potential of PhC sensors in the fields of neurology and medical diagnostics [127].

In the most recent study by Nazmi A. Mohammed et al. (2023), the authors made substantial progress. Their PhC sensors showcased remarkable advancements, boasting significantly enhanced sensitivity outcomes in comparison to their predecessors. These sensors introduced reduced sizes and streamlined fabrication processes, making them highly effective in detecting a broad spectrum of brain tissues and tumors. The detection limits for these sensors ranged from 10^−5^ to 10^−6^ RIU, promising great potential in the realms of medical diagnostics and neurological research [128].

Fluorescent peptide biosensors are sophisticated tools in molecular and cell biology research, tailored to monitor specific biological processes and interactions. These biosensors typically comprise a peptide sequence capable of undergoing changes in fluorescence in response to specific molecular events, such as enzyme activity. Through the detection and quantification of fluorescence changes, these biosensors offer invaluable insights into the dynamic nature of the biological processes they target. For instance, when designed to investigate kinase activity, a fluorescent peptide biosensor may contain a peptide sequence susceptible to phosphorylation by the kinase of interest. Phosphorylation alters the biosensor’s fluorescence properties, enabling the measurement of kinase activity. These biosensors prove instrumental in the real-time monitoring of cellular processes, readily applicable for live-cell imaging and a range of biological assays [129].

In the study by Peyressatre et al. (2020), an innovative fluorescent peptide biosensor took center stage, with a special focus on its application in glioblastoma research. This meticulously crafted biosensor was subject to rigorous validation, ensuring it provided a highly sensitive and selective quantification of CDK5 kinase activity. Importantly, it equipped researchers with the means to explore the intricate signaling pathways at play within glioblastomas, shedding light on the potential hyperactivation of CDK5 and CDK4 in specific cell lines, like U87, or in tumor biopsies. Moreover, the biosensor exhibited dynamic responses within living cells, particularly when stimulated by retinoic acid, ionomycin treatment, or cell starvation. This dynamic capability proved invaluable for assessing CDK5 activity in a physiologically relevant environment. The implications of this innovation are far-reaching, as it promises to significantly enhance our understanding of CDK5 dynamics in glioblastomas and its role in disease progression, potentially leading to more effective diagnostic and therapeutic strategies for this challenging medical condition [130].

In the 2016 study conducted by Karki et al., a significant breakthrough in enhancing the precision and specificity of glioma detection was achieved through the integration of optical biosensors and advanced nanomaterials. A remarkable nano-sized imaging agent with dual functionality was successfully developed, employing a G5 PAMAM dendrimer as a carrier for clinically relevant Gd-DOTA, a contrast agent used in MRIs, in conjunction with a fluorescent dye. Upon the systemic administration of (GdDOTA)54-G5-DL680, the agent demonstrated an exceptional capacity to selectively target glioma tumor sites. In vivo MRI scans clearly identified the presence of the agent in the glioma tumor, while no such signal was detected in the adjacent healthy tissue, underscoring its impressive specificity. To further corroborate its precision, whole-body NIR-optical imaging and ex vivo fluorescence imaging provided compelling evidence of the agent’s preferential localization. Notably, this dual-mode imaging agent exhibited remarkable versatility, with the MRI pinpointing the tumor’s macroscopic location and fluorescence imaging offering valuable insights into the agent’s biodistribution. This breakthrough underscores the practical applicability of the dual-mode imaging agent, holding great promise in the realms of medical imaging and glioma diagnosis [131].

Furthermore, surface plasmon resonance (SPR) emerges as an indispensable and versatile technology. As a label-free, real-time optical biosensing technique, SPR forms the cornerstone of investigations of molecular interactions and biomolecule detection. Its pivotal role in unraveling the intricacies of glioma biology is highlighted by its applications, including probing protein–protein interactions, enabling the early detection of critical biomarkers for disease monitoring, expediting drug development, shedding light on the role of exosomes in glioma progression, and facilitating personalized treatment strategies. With the ongoing enhancements of SPR biosensors to increase their sensitivity and specificity, this technology remains at the forefront of efforts to detect and understand glioma-related biomolecules with increasing precision [132].

In a 2018 study led by Qiu et al., an innovative biosensor named BAF-TiN was introduced as a precise and efficient tool for quantifying glioma-derived exosomes, with a specific focus on the CD63 marker and the glioma-associated protein EGFRvIII, which plays a pivotal role in glioma diagnosis. This groundbreaking biosensor provided a label-free, real-time, and highly sensitive method for quantifying these exosomes. The research team identified a biotinylated antibody capable of directly functionalizing a TiN nanofilm through TiN–biotin interactions. The immobilized biotinylated anti-CD63 and anti-EGFRvIII antibodies were utilized as highly sensitive and specific receptors for exosomes derived from U251 GMs, demonstrating the potential of plasmonic TiN biosensing. The BAF-TiN biosensor exhibited an impressively low limit of detection (LOD) for CD63 and EGFRvIII, making it a robust diagnostic tool. It not only demonstrated exceptional sensitivity but also displayed remarkable selectivity for U251GMs-derived exosomes, surpassing conventional SPR Au-film biosensors in various aspects. Notably, the BAF-TiN biosensors showcased their potential for quantifying exosomes and exosomal proteins in real biological fluids, extending their applicability to various biomedical fields, including disease diagnosis, immunotherapy, pathogen detection, and the pharmaceutical industry. Their biocompatibility and chemical stability further present the potential for real-time in vivo biotarget monitoring [133].

In summary, optical biosensors represent a rapidly advancing field that presents substantial potential for the diagnosis and treatment of gliomas, and the ongoing research and advancements in this area present the opportunities of improved patient care and decision support for this challenging medical condition.

## 3. Biotechnology Tools

### 3.1. Drug Delivery System

Nanocarriers, ranging in size from 1 to 100 nanometers, have emerged as a promising modality for enhancing drug delivery in oncology, presenting the potential to mitigate the systemic toxicity often associated with high concentrations of therapeutic agents. This approach is particularly advantageous in the context of glioma treatment. The versatile nature of nanocarriers allows for their utilization in various therapeutic domains, including chemotherapy, gene therapy, and immunotherapy, often with minor adjustments to the carrier platform. Within this framework, a spectrum of smart biomaterials is harnessed, including lipid carriers, polymer nanoparticles (NPs), metal nanoparticles (MNPs), bio-based NPs, and injectable or implantable 3D scaffolds. Each of these biomaterials serves as a unique carrier system, contributing to efficient drug encapsulation, customizable drug release, and targeted drug delivery. Additionally, nanocarrier systems enable the controlled release of therapeutic agents in response to external stimuli, such as changes in pH, mechanical forces, electrical signals, magnetic fields, light exposure, or variations in thermal conditions [134]. Furthermore, the incorporation of membrane-coated NPs with cell-mimicking attributes helps to circumvent interactions with immune cells in reticuloendothelial system (RESs), reducing the risk of phagocytosis and prolonging circulation in the bloodstream [114]. Moreover, the embedding of NPs in heat-sensitive hydrogels enhances site-specific retention within tumor tissues. These advancements are paramount for enabling sustained and controlled cargo release at the tumor site (Figure 4) [135].

Various targets have been identified for the precise targeting of glioma cells, with the majority of these targets being predominantly expressed by glioma cells. These include the low-density lipoprotein receptor (LDLR), EGFR receptors, mesenchymal–epithelial transition factor (MET), transferrin, and HER2/EGFR-tagged/decorated NPs, all of which are believed to facilitate the localization of nanoparticles (NPs) at the glioma injury site [136]. To further enhance targeting efficacy, it is recommended that the nanocarriers be coated with peptides or antibodies specifically designed to bind to brain endothelial cell (BEC) receptors involved in receptor transcytosis [137]. This targeted approach has great potential for improving the precision and efficiency of drug delivery to glioma cells (Figure 4).

### 3.2. Gene Circuits

The intricate web of molecular drivers and the inherent variability within and between gliomas have prompted the quest for innovative and tailored therapeutic strategies. In this pursuit, the integration of multicolored RNA circuits, designed to generate tumor suppressor microRNAs targeting pivotal glioma driver genes, stands as a pioneering approach. Several investigations have illuminated the therapeutic potential of specific miRNAs in the challenging landscape of GBMs. Notably, the overexpression of miR-25 and miR-32 has demonstrated the capacity to impede the growth of glioma stem cells, particularly when accompanied by low p53 signaling, underscoring their tumor-suppressive attributes [138]. miR-296-5p, on the other hand, has been observed to curtail the stemness of GBM cells, thereby diminishing their self-regenerative abilities [139]. Innovative delivery methodologies, such as dendritic polyglycerol amine (dPG-NH2), have facilitated the transportation of miR-34a across the blood–brain barrier, leading to the inhibition of GBM cell activities [140]. The augmentation of miR-378 expression has shown promise in enhancing GBM’s responsiveness to radiotherapy, presenting a potential avenue for therapeutic intervention [141].

miRNAs exhibit a remarkable capability to target multiple genes within intricate cellular pathways, rendering them attractive candidates for combination therapies. For instance, the utilization of polymeric nanogels to deliver NG-miR-34a nano-polyplexes in murine models has yielded encouraging results by effectively suppressing GBM cell proliferation [142]. Moreover, the upregulation of miR-139-3p has been identified as a potent inhibitor of GBM growth through targeted gene regulation, thereby warranting the consideration for therapeutic applications [143]. These investigations have also explored the utility of extracellular vesicles as vehicles for the delivery of miRNAs, rendering GBM cells more susceptible to treatment [144]. Furthermore, the deployment of CRISPR-Cas12a technology for the knockout of miR-21 has exhibited substantial promise in curbing GBM progression and enhancing overall survival [145]. miR-155 has emerged as a key modulator of angiogenesis, effectively mitigating GBM growth [146]. Lastly, the potential for combined therapies, involving specific miRNAs and adjunctive compounds, such as isothiocyanate sulforaphane and peptide nucleic acids, has been proposed to augment the efficacy of anti-GBM regimens while concurrently mitigating adverse side effects [147].

The concept of synthetic biological circuits holds great promise for inspiring future research directed toward targeting glioblastomas (GBMs) and preventing their recurrence. A study involving the analyses of the genomic, proteomic, post-translational modifications, and metabolomic data of 99 GBM patients has provided valuable insights into GBM biology. Phosphorylated PTPN11 and PLCG1 are considered potential switches that mediate oncogenic pathway activation and represent potential targets, particularly in EGFR-, TP53-, and RB1-altered tumors [148]. Synthetic gene circuits or sensor systems may be employed in conjunction with miRNA network approaches. In a study by Simion et al., an miRNA-ON monitoring system integrated into a lentiviral expression system (LentiRILES) served as an miRNA sensor system in mouse models of various cancer types, including GBMs, to monitor miRNA activities in single cells and track miRNA-based treatments [149]. Another group proposed that miR-1983 stimulated TLR7, which in turn induced the secretion of IFN-β, subsequently triggering the release of natural killer cells to target gliomas. This suggests that creating this innate circuit may pave the way for successful immunotherapy outcomes for GBMs [150]. Studies focused on innate regulatory circuits are crucial for enhancing our understanding of oncogenesis, GBM growth, and proliferation, ultimately inspiring the construction of synthetic circuits to inhibit GBM growth and prevent its recurrence. The regulatory circuit TCF4-miR-125b/miR-20b-FZD6 plays a crucial role in controlling the GBM phenotype innately. The insights from these studies provide a foundation for future research aiming to construct circuits to prevent the transition of GBM subtypes from PN to MES. miR-125b and miR-20b inhibit APC and FZD6, thereby enhancing Wnt signaling and inhibiting the generation of the MES subtype. Given the limited current research focusing on synthetic biological circuits for GBMs, it is essential that future investigations explore the potential of tools, such as CRISPR-Cas systems and RNA-based techniques, for targeting gliomagenesis, GBM maintenance, and growth-associated pathways [151].

Accordingly, multicolored RNA circuits, which are designed to generate tumor suppressor microRNAs targeting critical glioma driver genes, can be incorporated into nanohydrogels. These nanohydrogels consist of lipid gold nanoparticles coated with PLGA/PEG hydrogels and are tailored for local glioma therapy, effectively synthesizing the logical genetic circuits. This innovative approach empowers researchers to readily detect and quantify tumor heterogeneity by evaluating the treatment outcomes for each cell type constituting the tumor microenvironment. To assess the therapeutic efficacy on a cell-by-cell basis, multicolored microRNA circuits can be strategically expressed within distinct cell types present in the tumor microenvironment. These cell types include cancer cells, normal cells, immune cells, tumor-associated fibroblasts, endothelial cells, and tumor stem cells, and their specific expressions are facilitated through the use of cell-type-specific promoters. As a result, it becomes feasible to comprehensively assess tumor heterogeneity among diverse glioma cells, acknowledging the considerable degree of variability within and between tumors, as well as across individuals with gliomas. Furthermore, this approach enables the prediction of disease progression and the likelihood of therapy resistance. To bridge the translational divide between the preclinical and clinical outcomes, novel deregulated miRNA targets based on screenings conducted on patient-derived tumors provide a more accurate representation of distinct tumor microenvironments. This, in turn, enables the fine-tuning of both the hardware platform and genetic circuitry, as well as the assessment of the platform’s effectiveness in a patient-by-patient context using xenograft models derived from glioma patients. These approaches hold significant promise for advancing glioma therapy and personalizing treatment strategies.

The application of smart material platforms, in conjunction with newly identified biological targets, is promising for the development of effective cancer treatments, not limited to gliomas alone. Synthetic cancer-sensing circuits have been engineered to distinguish cancer cells based on intracellular gene expression profiles. The challenge of applying these circuits to cancer cells has, to date, constrained their efficacy, primarily leading to coupling with intracellular apoptotic mechanisms, thereby limiting their effectiveness against tumors. Since the majority of existing synthetic circuits are reliant on transcriptional regulation, the availability of synthetic transcription factors exhibiting both high efficiencies, capable of regulating target genes to the desired level of expression, and programmability is paramount for the construction of intricate circuits that can act upon user-defined target sequences.

Multi-output circuits designed to be specific to cancer and cell types exhibit the capacity to discriminate between human glioma cells and normal cells. These circuits integrate the activities of multiple synthetic promoters, including both glioma-specific and tumor-cell-specific promoters. When both types of promoters are active and found in both cancer and normal cells, they result in the low expression of heterologous proteins in normal cells and a high expression in glioma cells. This circuit architecture can initiate the generation of potent therapeutic effects in glioma cells by regulating the expression of selected miRNAs in the tumor microenvironment.

A comprehensive analysis of miRNA target interactions in the miRTarBase database has revealed nearly 1000 confirmed interactions, with 61% of the 59 target genes within glioma cancer stem cell extracellular vesicles (CSCEVs) being associated with pro-tumor genes and 25 with tumor suppressor genes. The subsequent screening of patient-derived cells has unveiled additional modified miRNAs that can facilitate the requisite material changes for the elimination of malignant tumors. The discovery of this multi-output barcode offers a highly efficient and selective tool for mapping the efficacy of miRNA therapy at a cellular level and for profiling tumor heterogeneity across diverse glioma neoplasms.

The gene circuits can be loaded into a specific viral vector, which is subsequently incorporated into a tailored scaffold responsive to various actuation mechanisms. Upon stimulation, viral particles carrying the gene circuits can be released in proximity to glioma cells, facilitating the delivery of these gene circuits into various glioma cell subtypes, ultimately leading to their inhibition or destruction.

In conclusion, the integration of innovative therapeutic strategies, such as multicolored RNA circuits, miRNA-based therapies, synthetic biological circuits, and advanced delivery methods, holds great promise for addressing the complexities of glioblastomas (GBMs). These approaches, as discussed, offer exciting opportunities to target critical glioma driver genes and enhance the tumor-suppressive potential of miRNAs, providing a multifaceted approach to tackling GBMs’ challenges. Furthermore, the ability to harness synthetic and innate regulatory circuits offers the potential for a more personalized and effective treatment of this devastating disease. In the future, it is crucial to continue exploring the translational potential of these cutting-edge technologies. Research efforts should focus on the fine-tuning and optimization of these platforms, including the identification of additional miRNA targets, which can be tailored to specific patient profiles. Moreover, clinical trials and in vivo studies are needed to validate the effectiveness and safety of these innovative therapies.

### 3.3. Fcγ-CR T-Cell Immunotherapy

Immunotherapies represent promising avenues for the treatment of glioblastomas (GBMs). These therapies encompass a wide range of approaches, including the targeting of variant epidermal growth factor III (EGFRvIII) and human epidermal growth factor receptor 2 (HER2) using engineered T cells, dendritic cell-based multi-peptides, dendritic cell-based vaccines, tumor-cell-generated vaccines, and natural killer cells [152]. Checkpoint inhibitors, microRNA, DNA, RNA, and viral vectors carrying specific genes or gene circuits are also under investigation for their potential in inhibiting and destroying glioma cells.

To achieve targeted immune cell or cancer cell interactions, nanoparticles can be modified with different ligands on their surfaces. These nanoparticles can respond to various multimodal stimuli, enhancing the delivery of therapeutic agents into the cytosol and enabling the sustained release of therapies. Several clinical studies have demonstrated the safety and efficacy of adoptive T-cell transfer for GBM immunotherapy. However, maintaining continuous T-cell activity within the tumor microenvironment (TME) remains a challenge.

Different types of chimeric antigen receptor (CAR) T cells have shown great promise against gliomas. Fcγ-chimeric receptor (CR) T cells (Fcγ-CR T cells) are a novel version designed to target a broader range of glioma-associated antigens [153]. Fcγ-CR T cells share transmembrane (TM) and intracellular chimeric signaling domains with conventional CAR T cells (Fc CR). Fcγ-CRs express the extracellular component of FcRs, whereas CARs include a single-chain variable fragment (ScFv) specific for tumor surface markers [153,154]. Fcγ-CR T-cell immunotherapy aims to transfer antibody-dependent cellular cytotoxicity (ADCC) activity from innate immune cells, such as natural killer (NK) cells, to T cells (Figure 5) [155].

The rationale for using Fcγ-CR T cells instead of NK cells is based on several factors: (1) T cells can be readily cultured in vivo and infiltrate the TME; (2) T-cell infiltration into tumors is generally associated with a positive prognosis [156]; and (3) NK cells may undergo apoptosis and experience the downregulation of CD16 and NK cell-activating receptors following a conjugation with tumor cells [157]. The role of NK cells in solid tumors is somewhat uncertain, as they often exhibit poor TME infiltration behavior and may not be directly correlated with positive prognoses [158].

Fcγ-CR T lymphocytes, equipped with monoclonal antibodies, can be highly specific against surface antigens associated with gliomas, such as epidermal growth factor receptor (EGFR) and platelet-derived growth factor (PDGFA). These innovative immunotherapeutic approaches present significant potential for advancing the treatment of GBMs.

In conclusion, immunotherapies offer a promising outcome in the battle against glioblastomas (GBMs). Diverse strategies, including engineered T cells, dendritic cell-based vaccines, and nanoparticles with tailored ligands, are being explored to enhance the targeting and treatment of GBMs. Chimeric antigen receptor (CAR) T cells, particularly novel Fcγ receptor (CR) T cells, show great potential for extending the scope of glioma-associated antigen targeting. This innovative approach holds significant promise for advancing GBM treatment by harnessing the capabilities of engineered immune cells and nanotechnology, thus providing hope for improved outcomes in the fight against this formidable disease.

### 3.4. Targeted Treatment, Cytokine Release Syndrome Management, and Advanced Nanoplatform Systems

Serious side effects are commonly associated with systemic chemotherapy and radiotherapy. Additionally, the utilization of adoptive cell therapy (ACT), including CAR-T-cell immunotherapy, can result in the severe and often catastrophic occurrence of cytokine release syndrome (CRS). This syndrome is characterized by the excessive release of cytokines, leading to a systemic inflammatory response with potentially life-threatening consequences [159].

The controlled release of various therapeutic agents, including small molecules, chemotherapy drugs, radiotherapy, viral vectors carrying inhibitory genes, gene circuits, microRNAs, CAR T cells, genome-editing components, and more, can be achieved through the interface of a nanoplatform system (NPS) strategically positioned alongside the glioma microenvironment. This release can be initiated by nanomaterial-based cross-modal modulations, encompassing electrical, mechanical, magnetic, optical, thermal, and optogenetic mechanisms, all managed by an internal chip (Figure 6) [160,161,162].

This pioneering approach establishes a closed-loop system for sensing and actuation, allowing for the precise tailoring of therapeutic interventions based on the levels of cytokine release within the tumor microenvironment (TME). This dynamic feedback mechanism significantly augments the efficacy of the proposed multimodal therapy customized for gliomas, presenting a promising strategy for enhancing treatment outcomes.

Furthermore, owing to the reactivity of hybrid structures, input signals can be channeled to the glioma microenvironment through previously unexplored pathways. These pathways encompass overcoming challenges, such as tissue scatter in visible-light transmission, or directly altering membrane polarity at the position of the electrode. This capacity to explore novel avenues for signal transmission underscores the potential for significant advancements in the field of glioma therapy.

## 4. Bioelectronic Sensors

### 4.1. Nanomaterials as an Interface for Targeting Gliomas

A new bioelectronic system has been created that combines genetic and inorganic materials with various physical properties, and plasmonic and magnetoelectric nanoparticles have recently been employed as stimulation interfaces in brain regions [163]. When plasmonic nanoparticles are exposed to light, they can be induced to emit heat. However, as a light source must be implanted to function in vivo, these technologies have primarily been utilized in vitro [164]. Magnetoelectric particles (piezoelectric ceramics) are used for transgene-free activation, but their clinical application is limited due to the potential release of neurotoxic substances, such as barium (Ba) and cobalt (Co), upon decay [165].

Nanoscale materials exhibit size-dependent characteristics, quantum confinement, a high surface-to-volume ratio, and increased catalytic activity. Hydrogels and polymers are used to reduce the surface modulus of electrodes, enabling them to conform to tissues and prevent chronic glial scarring. These materials can also respond to environmental changes, including variations in temperature, pH, and electrical impulses. Utilizing hydrogels and actuators based on nanomaterials facilitates the creation of highly sensitive sensors with robustness and rapid response times. Polymeric materials, including viral vectors [166], have been developed to sensitize glioma cells to ion channels thermally, mechanically, or optically, enabling precise spatial targeting of glioma cells [167]. This approach allows the monitoring of chemical changes in the local environment when thermal, magnetic, mechanical, electrical, or optogenetic stimuli are applied. The use of viral vectors containing genetic material streamlines one-step transfection and sensitization of glioma cells to external stimulation, eliminating the need for many implantation procedures.

Targeted drug delivery systems have demonstrated significant advantages by leveraging both inorganic and organic nanomaterials. When considering non-invasive actuation interfaces within brain tissue, it is imperative that nanomaterials possess elastic properties comparable to those of the brain itself. Plasmonic gold nanoparticles (GNPs) have been employed in the context of photothermal treatment (PTT), primarily through the absorption of intense light sources, such as lasers. During this intricate process, electron transfer mechanisms play a vital role in facilitating the transmission of heat generated within the crystal lattice to the surrounding environment. This phenomenon is observed as the laser liberates free electrons from the NP plasmon band [168]. Nano stars (multi-branched asymmetric Au NPs) and nanorods (stretched Au NPs in one direction) can locally increase temperature by exciting plasmonic bands using 630–650 nm light from a microchip µLED. When the NP surface is illuminated, the “nanoantenna” phenomenon creates a field amplification [137,152]. Tuned Au NPs become highly efficient local heaters, generating “hot spots” that can elevate local temperatures by approximately 3 degrees Celsius and activate heat-sensitive ion channels [169]. For practical implementation, Au NPs with a low aspect ratio and an excitable plasmon resonance are excellent candidates for electrode construction [170]. Among metallic nanoparticles, iron oxide nanoparticles (IONPs) have received FDA approval for clinical use for cancer treatment [171].

Gold nanoparticles, as well as the size and shape of the generated nanorods, can be adjusted to enhance the conversion of light into heat, aligning with the emission wavelength of the microchip. Polymers are composed of covalently bonded subunits that determine their properties. Core–shell multiferroics and custom coatings of CoFe_2_O_4_–BaTiO_3_ polymers for use in the microchip were synthesized via hypothermic methods on the electrode’s surface. To provide effective mechanical stimulation, modifications were made to the core, shell, and size. Layer-by-layer (LbL) techniques have been employed for electrode coating. Polyelectrolytes, such as polyallylamine hydrochloride (PAH), polyacrylic acid (PAA), or poly(3,4-ethylenedioxythiophene) PEDOTN, have been used to enhance electrical contact [171,172].

Electrodes coated with amino-terminated polyethylene glycol can improve their biocompatibility and allow for electrostatic assembly. For viral transmission, hydrogel particles loaded with polycation poly(2-(diethylamino) ethyl methacrylate) (PDEA) or poly(methacrylic acid) polyanion (PAA) can be coated onto the electrodes through free radical polymerization [173,174]. Adjusting the hydrogel’s porosity, size, and density of functional groups can maximize the reactivity of the hydrogel [175].

The convergence of nanomaterials and bioelectronic systems presents an exciting discovery in the battle against gliomas. This multidisciplinary approach, incorporating genetic and inorganic materials, reveals the potential to revolutionize the precision and effectiveness of glioma diagnosis and treatment. By harnessing the unique properties of nanomaterials and their compatibility with a range of physical properties, there is an opportunity to develop innovative solutions that can minimize the harm to healthy brain tissues.

The application of nanomaterials as targeting interfaces presents an opportunity for the creation of more personalized and less invasive treatments for gliomas. Continuing to explore and refine these approaches, the future holds great promise for improved outcomes in the fight against this formidable brain cancer. Research in this field is vital and presents opportunities for the development of advanced therapies and diagnostics that can positively impact the lives of glioma patients.

### 4.2. Ultraminiaturized, Wirelessly Charged, and Biocompatible Implantable Electronics

In the field of medical technology, significant attention is being directed toward the development of ultraminiaturized implantable electronics. These devices are wirelessly charged and designed to be biocompatible. These pioneering technologies have the potential to revolutionize in the fields of medical devices and diagnostics. Their discovery introduces a set of distinctive capabilities that can significantly enhance the precision and minimally invasive nature of medical interventions.

In recent soft and flexible photoelectric systems, ultraminiature, wirelessly driven, and controlled micro-luminescent diodes (µLEDs) can deliver light directly to areas of interest. Additionally, power transmission depends on a bulky external motion-sensing device. Despite minimal diffraction and signal security, several obstacles remain, including complicated electronics, the difficulty of targeting proper frequencies, poor communication speeds, and noticeable skull reduction. Due to the lack of programmable control and limited lighting profile options, the existing fixtures are limited in their applicability [176].

The additional functionalities of the implantable microchip are designed to achieve long-term tissue compatibility [177]. This is achieved through full encapsulation using ultrathin, biocompatible polymers, such as poly(ethyl acrylate). The external device is a metamaterial-based tunable broadband transmitter, enhancing wireless power transfer (WPT) reliability, operational range, and displacement tolerance. Furthermore, a sound-activated magnetoelectric (ME) antenna is employed to increase the signal capture in comparison to traditional electromagnetic (EM) antennas, while simultaneously minimizing signal attenuation through the skull.

The small biocompatible micro-device serves as a wireless integrated tissue transducer. It possesses the capability to sense factors like cytokine levels and activate various therapeutic approaches targeting glioma cells across different regions of the brain. This multifunctional device enables the real-time monitoring of chemical changes in the local environment, responding to thermal, magnetic, mechanical, electrical, or optogenetic stimuli [178] (Figure 6).

In a study conducted by Pierpaolo Peruzzi in 2023, it was demonstrated that drug-releasing intra-tumoral microdevices (IMDs) could be safely and effectively utilized to acquire the patient-specific, high-throughput molecular and histopathological profiles of drug responses in gliomas. These findings offer groundbreaking evidence for their first-in-human use and their potential to complement other strategies for selecting drugs based on their observed antitumor effects in the tumor’s natural environment.

IMDs are seamlessly integrated into the surgical process during tumor resections and are left in place only for the duration of the standard operation, which typically lasts for 2 to 3 h. Remarkably, none of the six enrolled patients experienced adverse events related to the IMDs, and the collected tissue was suitable for downstream analyses for 11 out of 12 retrieved specimens. The analysis of these specimens yielded preliminary evidence of the reliability of the data obtained, their compatibility with a wide range of techniques for molecular tissue analysis, and promising correlations with the observed clinical and radiological responses to temozolomide. From an investigational perspective, the wealth of information obtained using IMDs enabled the characterization of tissue responses to various drugs of interest within the tumor’s natural context, all without disrupting the standard surgical workflow [179].

As we continue to explore new avenues in glioma research, there is reason to be optimistic about the future of glioma treatment. By combining personalized medicine approaches with cutting-edge technologies, like IMDs, we are moving closer to a time when glioma patients may have access to more effective and tailored therapies, ultimately improving their quality of life and prognosis. The journey to conquering gliomas is ongoing, but the recent advancements offer hope for a brighter future in the fight against this devastating disease.

### 4.3. Neuromorphic and Memristive Computing

Neuromorphic computing has emerged as one of the most promising models for overcoming the limitations of the von Neumann design in standard digital processors [180]. Current neuromorphic devices make use of metal oxide semiconductor (CMOS) technology. However, CMOS technology has reached its physical limits, with limited material selection and manufacturing processes that can only obtain the smallest feature sizes possible due to the demand for ever-smaller and more portable electronic devices. This situation gives rise to several significant issues, including high power losses, low reliability, thermal impacts, and the inability to shrink the technology further [181]. Digital CMOS approaches necessitate many clock cycles (resulting in high latency) to perform multiplication and accumulation operations. Continuous power is required because network parameters are stored in volatile elements, like SRAM.

The introduction of a glioma chip-integrated neuromorphic computing approach offers the potential to address multiple constraints at once, presenting several clear advantages:Low power consumption: this solution is characterized by low power consumption, ensuring energy efficiency.Intrinsic non-volatile memory: the system’s intrinsic non-volatile memory enables the encoding of “artificial synaptic forces”, even in the absence of electrical power, preserving weight and parameter information.High scalability: the approach is highly scalable, allowing for the storage of multiple bits within a single device. This scalability enhances the storage density and conserves space, moving closer to the computing power-to-volume ratio observed in mammalian brains.Low latency: with fast write and read times, the system achieves very low latency results, resulting in a quicker system response. Additionally, it eliminates the physical separation between the computer and memory unit.Minimal electricity consumption: this approach consumes minimal electricity, further enhancing its energy efficiency.

Encouraging patients to engage in the adaptive, cross-modal activation of multimodal therapeutic techniques against glioma cells is essential. This approach allows for various activation methods, including electrical, mechanical, magnetic, optical, thermal, and optogenetic. This innovative glioma chip-integrated neuromorphic computing approach holds great promise in overcoming the existing constraints and advancing the treatment of gliomas [182].

The study by Trensch and Morrison, published in 2022, highlights the challenges of understanding brain function and the need for large-scale neural network simulations. These simulations in hyper-real-time are crucial for comprehensive parameter exploration and studying slower processes, like learning and memory. The paper points out that even the fastest supercomputers cannot perform such simulations accurately and reproducibly. While neuromorphic computer architectures are promising, their high costs and long development cycles hinder their ability to keep up with neuroscience advancements. To address this, the authors proposed a novel hybrid software–hardware architecture for a neuromorphic compute node designed to function in a multi-node cluster configuration. They based their design on the Xilinx Zynq-7000 SoC, which combined a potent programmable logic gate array (FPGA) with a dual-core ARM Cortex-A9 processor extension on a single chip. This architecture effectively utilizes both components, enabling the construction of smaller neuromorphic computing clusters capable of hyper-real-time simulations involving tens of thousands of neurons. This approach addresses the high demands for modeling and simulating neural networks in neuroscience [183].

Neuromorphic computing, with its concurrent processing and pattern recognition capabilities, offers valuable support for the early detection and categorization of gliomas through the analysis of medical imaging data. Memristive devices, renowned for their non-volatile memory attributes, play a crucial role in the efficient storage of large datasets, including patient records and medical imaging information. These technologies play a pivotal role in the formulation of machine learning algorithms for treatment planning, enabling dynamic adjustments to treatment strategies in real time. Furthermore, neuromorphic systems can replicate neural networks for the examination of treatment impacts, while memristive computing, featuring artificial synapse capabilities, facilitates an instantaneous adaptation to treatment outcomes. These computing paradigms enable the real-time monitoring of the local brain environment during therapy, offering the potential for personalized and adaptive treatment strategies. They also excel in the secure and efficient management of the copious multi-modal data generated by glioma patients, ultimately contributing to the augmentation of precision and efficacy in glioma diagnosis and treatment, all the while mitigating the harm to healthy brain tissues.

In conclusion, the fusion of neuromorphic computing and memristive devices represents a pivotal development in the fields of glioma research and treatment. These technologies, with their remarkable concurrent processing and pattern recognition capabilities, are poised to revolutionize the early detection and categorization of gliomas through the analysis of extensive medical imaging data. Memristive devices, revered for their non-volatile memory attributes, are instrumental for efficiently storing large datasets, including patient records and medical imaging information. What makes these technologies even more compelling is their role in the development of machine learning algorithms for treatment planning. They enable dynamic adjustments to treatment strategies in real time, ensuring that therapeutic interventions are tailored to the unique needs of each patient. Moreover, neuromorphic systems have the ability to replicate neural networks for the in-depth assessment of treatment impacts, while memristive computing, with its artificial synapse capabilities, enables an instantaneous adaptation to treatment outcomes.

This synergy of neuromorphic and memristive computing paradigms allows for the real-time monitoring of the local brain environment during therapy. It allows for the development of personalized and adaptive treatment strategies, optimizing precision and efficacy. In the era of big data generated by glioma patients, these technologies excel in securely and efficiently managing the copious amount of multi-modal data. Their implementation holds the promise of revolutionizing glioma diagnosis and treatment, while minimizing the harm to healthy brain tissues.

### 4.4. Multimodal, Multi-Site, and Adaptive in-Brain Glioma Therapeutics

Reactive transcranial techniques have been previously documented, predominantly encompassing single-site, single-intervention modalities, which are responsive to electrocorticography (EEG) patterns for the treatment of brain diseases. However, these techniques exhibit limitations in their capacity to investigate multiple anatomical sites, target and modulate multiple neural loci, and respond to the unique pathophysiological patterns of individual users [184].

The anticipated advancements in this field propose the development of integrated AI-based approaches to facilitate the intercommunication among unified and networked implanted microchips, allowing for the coordinated implementation of therapeutic interventions. Additionally, these advancements may include the deployment of multiple networked microchips, each coated with advanced nanomaterials functioning as actuators, with each microchip presenting the capability to perform local sensing and provide localized multimodal stimulations. These advanced microchips are envisioned to possess adaptive and self-learning properties, characterizing their functions and attributes based on the contributions offered by the network of microchips positioned in proximity to glioma lesions.

The development and exploration of such microchips involve several critical facets, including microchip design, biosensing technologies, the efficiency of memristive devices, the assessment of glioma lesion progression, and a range of methodologies for integrating microchips into brain tissue, including in vitro and in vivo implementations [185,186,187].

### 4.5. Microchip Hardware Development

The hardware development for a microchip designed to combat glioblastomas, an aggressive form of brain cancer, represents a significant advancement in medical technology. These microchips are equipped with cutting-edge features to enable wireless power transfer and present advanced communication capabilities. A dual-band RF antenna, designed for communication and wireless power transfer, ensures efficient multitasking. To guarantee safety and flexibility for implantation in the brain, a small magnetoelectric (ME) antenna combined with a biocompatible polymer binder was employed. Extensive simulations using Multiphysics software, such as, IBM Neural Computer INC-3000, Hybrid neuromorphic compute (HNC) node and System-on-Chip (SoC) devices in a high bandwidth 3D mesh communication networkcan fine-tun the antenna’s resonant frequency, resulting in its compact size, ideal for brain implantation [183].

The development also includes a read/power management interface chip, essential for facilitating wireless power and data operations. These chips streamline the flow of power and data between the microchip and external devices, ensuring consistent and efficient operations. To maintain a reliable power supply, the voltage received by the multiband antenna is rectified using RF Schottky diodes and regulated by a low-dropout voltage regulator. In cases of power fluctuations, a supercapacitor temporarily stores energy, ensuring an uninterrupted operation.

Moreover, the multi-band antenna not only receives power, but also facilitates the wireless transmission of biological data from the microchip to a neuromorphic computer system. This ability is pivotal for the real-time monitoring and adjustment of glioblastoma treatment strategies. In essence, this hardware development promises a groundbreaking approach to tackling glioblastomas, offering a powerful, flexible, and precise solution for diagnosis and management. It holds the potential to improve patient outcomes and advance the fight against this devastating disease.

### 4.6. Cytokine Sensor Development

Developing cytokine sensors for monitoring cytokine release at glioma sites is a promising approach in the field of glioma treatment. This cutting-edge technology involves using microchips equipped with unique sensors that wirelessly transmit data to an external control unit, allowing for the real-time monitoring of cytokine levels. The goal is to gain insights into the cytotoxic activity of glioma cells by analyzing the cytokine release patterns. One crucial aspect of this approach is the utilization of array-grafted microarrays, which can observe specific subdomains of the array, allowing for the identification of cytokine release modulation. This enables tailored and multimodal stimulations of nanoparticle interfaces, triggering various therapeutic modalities. The ultimate objective is to restore abnormal cellular behavior to a more physiological phenotype.

The collected cytokine release data are wirelessly transmitted to a neuromorphic chip, a key component of the external control unit. Here, artificial intelligence (AI) algorithms are employed to differentiate between normal and glioma-associated cytokine release patterns. The data are then securely stored in a memristic memory chip, ensuring that the analysis can continue indefinitely. The external microchip plays a pivotal role in this approach as it can wirelessly activate different types of interventions based on the cytokine levels in the tumor microenvironment (TME). This precise control allows for a targeted and adaptive response to the specific needs of individual glioma lesions. Furthermore, the approach incorporates cutting-edge CRISPR-Cas9 gene editing technology to eliminate programmed cell death protein 1 (PD1) in glioma T cells. This genetic modification has the potential to enhance safety and reduce toxicity associated with autoimmune responses. By eliminating PD1, the immunological checkpoint molecule, the approach aims to mitigate the autoimmunity-induced toxicity that may promote protumor activity [188]. In the future, the aim is to expand the application of this technology by incorporating a cytokine detection sensor to measure the release of specific cytokines, such as IFNγ and TNFα. This will provide even more precise information about the tumor microenvironment, further enhancing the effectiveness of glioma treatment strategies.

## 5. Conclusions

In conclusion, the pursuit of theragnostic sensors for tackling gliomas represents a multifaceted and promising approach to addressing the challenges posed by this formidable brain tumor. Gliomas, marked by their aggressiveness, limited treatment options, and high recurrence rates, necessitate innovative approaches beyond the conventional diagnostic methods.

This review highlighted various cutting-edge technologies and biocompatible interfaces that show the potential of revolutionizing glioma treatment. Techniques such as optical coherence tomography (OCT), Raman-based spectroscopy, reflectometric interference spectroscopy (RIfS), and colorimetric methods offer enhanced precision and non-invasive capabilities for glioma diagnosis and monitoring. These advancements have the potential to streamline surgical procedures, improve diagnostic accuracy, and reduce the risks associated with traditional biopsy methods.

In the field of nanotechnology, the integration of nanocarrier-based drug delivery systems and advanced nanomaterials within bioelectronic systems presents new opportunities for more precise and personalized glioma therapies. Targeted drug delivery systems and innovative nanomaterials present the potential for more efficient treatments while minimizing systemic toxicity. Plasmonic nanoparticles, magnetoelectric particles, and stimulation interfaces in brain regions present new possibilities for addressing the complex nature of gliomas.

The incorporation of neuromorphic computing, memristive devices, and multimodal, multi-site, and adaptive in-brain glioma therapeutics has transformative potential for diagnostics and treatments. These technologies enable real-time monitoring, enhancing precision and efficacy, while minimizing damage to healthy brain tissue. Cytokine sensors provide valuable insights into the glioma site, facilitating remote analysis and informing treatment strategies.

In the realm of immunotherapies, the development of chimeric antigen receptor (CAR) T cells, particularly novel Fcγ receptor (CR) T cells, extends the scope of glioma-associated antigen targeting. These innovative approaches, leveraging engineered immune cells and nanotechnology, offer hope for more effective treatments and improved prognosis.

Collectively, these developments signify a comprehensive effort to treat gliomas by enhancing precision, improving diagnostic accuracy, and providing more personalized and effective therapeutic strategies. The ongoing research in these areas is optimistic concerning the fight against this challenging disease.

The future directions in the development of theragnostic sensors for treating gliomas have immense potential for further advancements in glioma diagnosis and treatment strategies. These directions encompass the integration of artificial intelligence (AI) to enhance precision, large-scale clinical trials to validate safety and effectiveness, personalized medicine, non-invasive monitoring, interdisciplinary collaboration, biomarker discovery, improved nanocarrier systems, safety and ethical considerations, patient education, and global collaboration. These collective efforts contribute to the prospect of better outcomes for glioma patients and, ultimately, the treatment of this formidable disease.

## Figures and Tables

**Figure 1 sensors-23-09842-f001:**
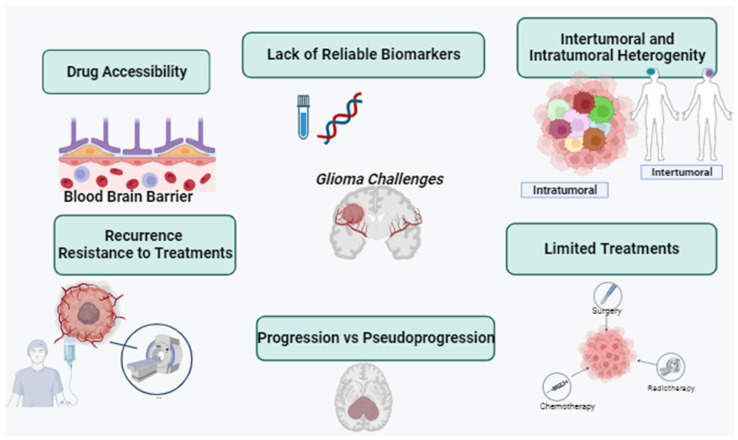
Challenges in glioma research, such as drug accessibility, lack of reliable biomarkers, recurrent resistance of the treatment, progression vs. pseudoprogression, and intertumoral and intratumoral heterogeneity (created with the free trail of BioRender).

**Figure 2 sensors-23-09842-f002:**
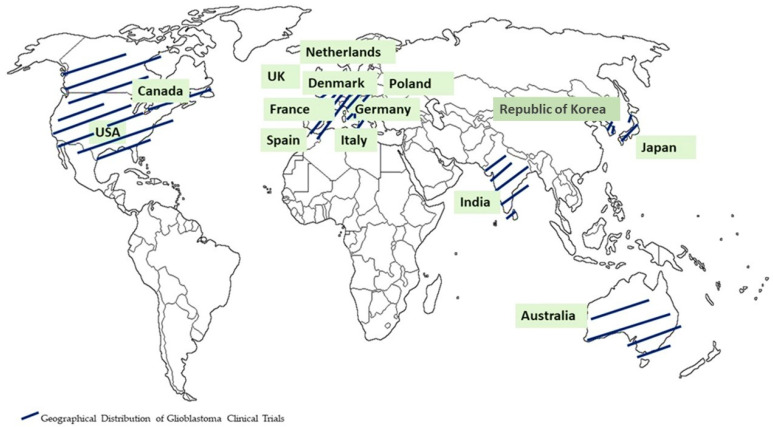
Geographical distribution of glioblastoma clinical trials.

**Figure 3 sensors-23-09842-f003:**
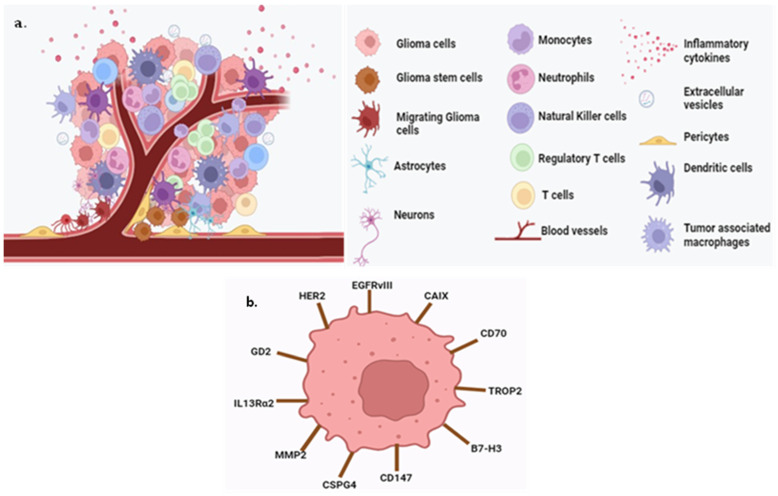
Scheme of the glioma tumor microenvironment and biomarkers. (**a**) Tumor-associated macrophages, regular T cells, monocytes, neutrophils, neurons, pericytes, glioma cells, inflammatory cytokines, astrocytes, extracellular vesicles, blood vessels, dendric cells, and NK cells are present in the glioma tumor microenvironment. (**b**) The biomarkers are B7-H3, CD70, CD147, CAIX, CSPG4, GD2, EGFRVIII, HER2, IL13Rα2, MMP2, and TROP2 (created with the free trail of BioRender).

**Figure 4 sensors-23-09842-f004:**
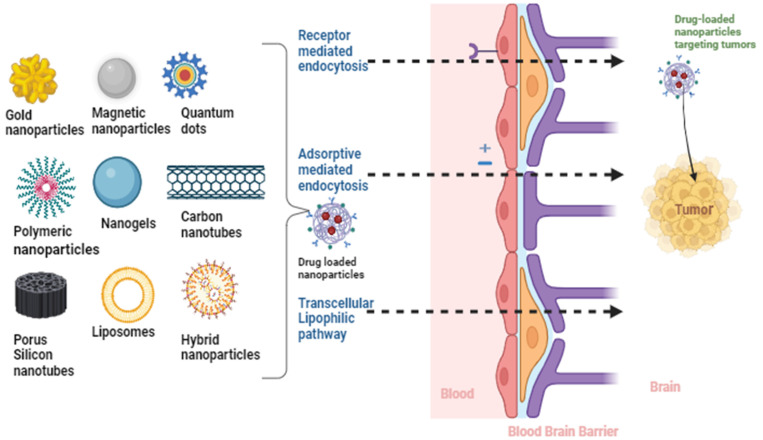
Nanomaterial-based targeting for glioma therapy with emphasis on overcoming the blood–brain barrier: the cutting-edge utilization of nanomaterials in the context of precision targeting and therapeutic interventions for gliomas (created with the free trail of BioRender).

**Figure 5 sensors-23-09842-f005:**
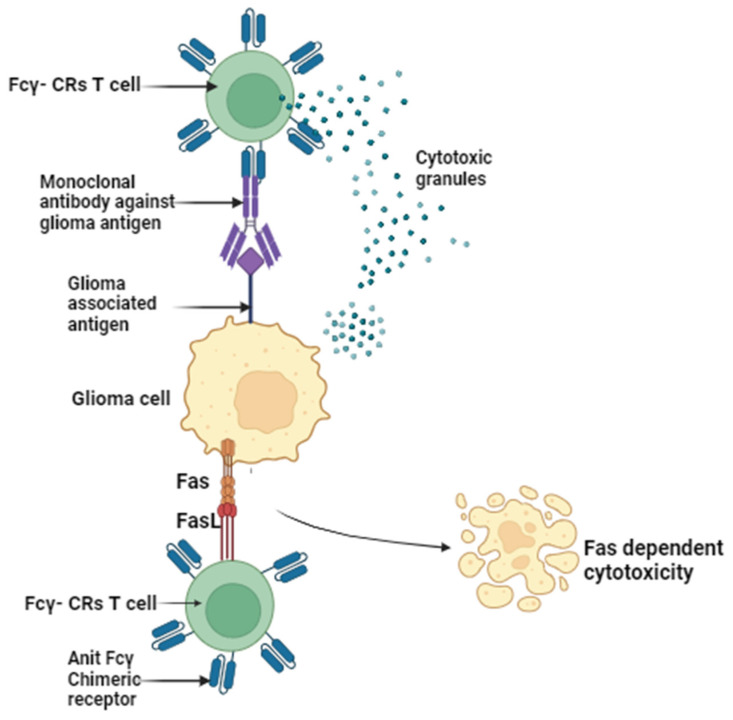
Elucidating the mechanisms of Fc gamma chimeric receptor cell-mediated (Fcγ-CRs T) eradication of tumor cells: the Fc fragment of monoclonal antibodies (mAb) binding to Fcγ-CRs to Fcγ-CR T cells. In response to the identification of glioma-associated antigens and the subsequent binding of monoclonal antibodies (mAb) to Fcγ-CRs T cells, the activation of Fcγ-CRs T cells ensues, resulting in the initiation of cell-mediated cytotoxicity through the release of cytotoxic granules or the activation of FAS expression dependent on Fcγ-CRs (created with the free trail of BioRender).

**Figure 6 sensors-23-09842-f006:**
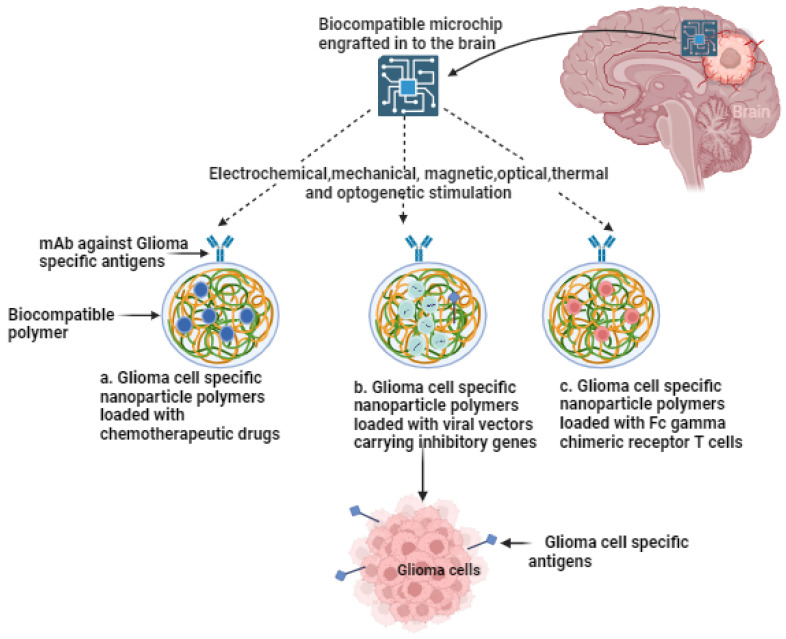
Multimodal stimulation of nanoparticles with an engrafted microchip for the controlled release of diverse therapeutic agents in the context of glioma therapy The versatile activation of nanoparticles to enable the controlled discharge of various therapeutic agents, including chemotherapeutic agents, viral vectors carrying inhibitory genes, Fcγ-CR T cells for targeted drug delivery, transfection of glioma cells, and the selective expression of ion channels. Integration of inhibitory gene circuits within glioma cells responsive to light, heat, or mechanical forces (created with the free trail of BioRender).

**Table 1 sensors-23-09842-t001:** Review of the clinical trials for glioblastoma treatment.

Treatment	Phase	Concentration/Dose	Sample Size	Result	Country	Year	Reference
Lomustine (CCNU)–temozolomide (TMZ) combination therapy vs. standard temozolomide	Phase III Open-label, randomized NCT01149109	Standard TMZ chemoradiotherapy: TMZ: 75 mg/m^2^ per day; Radiotherapy (59–60 Gy); Followed by six courses TMZ: Dose: 150–200 mg/m^2^ per day. Combination therapy (TMZ+CCNU): CCNU: 100 mg/m^2^ on day 1; TMZ: 100–200 mg/m^2^ per day; In addition to radiotherapy (59–60 Gy).	141 patients: TMZ: 63 TMZ+CCNU: 66	Median overall survival: TMZ: 31.4 months; TMZ+CCNU: 48.1 months. Significant overall survival difference in favor of lomustinetemozolomide. Larger studies are needed to validate the observed clinical implications.	Germany	2011–2014	[27]
Nivolumab monotherapy (NIVO) and combination with ipilimumab (IPI) for recurrent glioblastomas	Phase I NCT02017717	NIVO monotherapy (NIVO3): 3 mg/kg; Every 2 weeks. Combination therapy (NIVO1+IPI3): 1 mg/kg + 3 mg/kg; Every 3 weeks for 4 doses. Alternative combination therapy (NIVO3+IPI1): 3 mg/kg+1 mg/kg; Every 3 weeks for 4 doses.	40 patients: NIVO3: n = 10; NIVO1+IPI3: n = 10; NIVO3+IPI1: n = 20.	Median overall survival (OS): NIVO3 (10.4 months), NIVO1+IPI3 (9.2 months), and NIVO3+IPI1 (7.3 months). Nivolumab monotherapy showed a better tolerability than the combination therapy and was selected for a phase III cohort.	USA	2017	[28]
Dabrafenib and trametinib combination therapy in patients with BRAFV600E-mutant low-grade and high-grade gliomas	Phase II Basket trial Open-label, single-arm NCT02034110	Dabrafenib: 150 mg twice daily orally. Trametinib: 2 mg once daily orally.	45 patients: 31-high-grade glioma cohort; 13-low-grade glioma cohort.	Dabrafenib+ trametinib showed clinically meaningful activity. Effective in BRAFV600E mutation-positive recurrent or refractory high-grade and low-grade gliomas. Safety profile consistent with other indications. Suggests the potential adoption of BRAFV600E testing in clinical practice for glioma patients.	Part of an ongoing study, basket trial including 13 countries: Austria, Belgium, Canada, France, Germany, Italy, Japan, Netherlands, Norway, Republic of Korea, Spain, Sweden, and USA.	2014–2018	[29]
Selinexor monotherapy in recurrent glioblastomas	Phase II open-label study NCT01986348	Arm A (Pre-surgical selinexor): 3 preoperative selinexor doses. Median selinexor concentration in resected tumors: 105.4 nmol/L. Arms B, C, and D (post-operative selinexor): Arm B: selinexor 50 mg/m^2^ twice weekly. Arm C: selinexor 60 mg twice weekly. Arm D: selinexor 80 mg once weekly.	76 patients: Arm A: 8; Arm B: 24; Arm C: 14; Arm D: 30.	Progression-free survival (PFS6) at 6 months: Arm B: 10%; Arm C: 7.7%; Arm D: 17.2%. Median overall survival (OS): ranged from 8.5 to 10.5 months in arms B, C, and D. Overall response rate (ORR): varied from 7.7% to 10% across arms. Selinexor monotherapy (80 mg weekly): favorable responses and clinically significant 6-month PFS. Manageable side effects, even with dose reductions. Ongoing trials to evaluate the safety and efficacy of Selinexor in combination with other therapies.	USA; Denmark; Netherlands.	2020	[30]
Repetitive blood–brain barrier opening via implantable ultrasound device for albumin-bound paclitaxel delivery in recurrent glioblastomas	Phase I	Low-intensity pulsed ultrasound with concomitant intravenous microbubbles (LIPU-MBs) with intravenous albumin-bound paclitaxel infusion. Frequency: every 3 weeks. Duration: up to six cycles. Dose escalation: Six dose levels of albumin-bound paclitaxel were evaluated: 40 mg/m^2^; 80 mg/m^2^; 135 mg/m^2^; 175 mg/m^2^; 215 mg/m^2^; 260 mg/m^2^. Primary endpoint: The evaluation of the dose-limiting toxicity during the first cycle of sonication and chemotherapy	17 patients	Imaging analysis showed the blood–brain barrier opening in targeted brain regions, diminishing over the first 1 h after sonication. Pharmacokinetic analysis: LIPU-MB led to significant increases in the brain parenchymal concentrations of albumin-bound paclitaxel and carboplatin. The positive outcomes of this study led to the initiation of a follow-up phase 2 trial (NCT04528680).	USA	2020–2022	[31]
Mebendazole (MBZ) combination therapy with lomustine (CCNU) or temozolomide (TMZ) in recurrent glioblastomas	Phase II Randomized open-label trial CTRI/2018/01/011542	CCNU+MBZ (mebendazole) arm: CCNU: 110 mg/m^2^ every 6 weeks; MBZ: 800 mg thrice daily. TMZ+MBZ arm: TMZ: 200 mg/m^2^ once daily on days 1–5 of a 28-day cycle; MBZ: 1600 mg thrice daily.	44 patients randomized in each arm	The addition of mebendazole (MBZ) to temozolomide (TMZ) or lomustine (CCNU) did not attain the predefined benchmark of a 55% 9-month overall survival (OS). This outcome may be attributed to the presence of 28.6% of patients with a poor Eastern Cooperative Oncology Group Performance Status (ECOG PS) of 2–3.	India	2019–2021	[32]
Nivolumab vs. bevacizumab in patients with recurrent glioblastomas	Phase III Randomized open-label trial NCT02017717	Nivolumab: 3 mg/kg every 2 weeks; Bevacizumab: 10 mg/kg every 2 weeks.	369 patients: nivolumab: 184; bevacizumab: 185.	Lack of survival improvement with nivolumab compared to the control (bevacizumab). Further investigation is necessary. Ongoing research: exploring the efficacy of nivolumab in conjunction with radiotherapy and temozolomide patients with methylated MGMT promoter.	12 countries: USA; Australia; Belgium; Denmark; France; Germany; Italy; Netherlands; Poland; Spain; Switzerland; UK.	2014–2015	[33]
Triple-mutated oncolytic herpes virus G47∆ in patients with progressive glioblastomas	Phase I/II UMIN000002661	Cohort 1: G47Δ of 3 × 10^8^ pfu (total of 6 × 10^8^ pfu). Cohort 2 and the phase II part: G47Δ at a dose of 1 × 10^9^ pfu (total of 2 × 10^9^ pfu). Administration protocol: Total volume: 1 mL. G47∆ injected into two different sites. Each site received 0.5 mL.	13 patients: Cohort 1: 3; Cohort 2 and the phase II part: 10.	Revealed tumor cell destruction via viral replication. Showed features such as injection site contrast-enhancement clearing and entire tumor enlargement. Study concluded that G47Δ is safe for treating recurrent/progressive glioblastomas. Recommends further clinical development based on the observed outcomes.	Japan	2009–2014	[34]
Intratumoral oncolytic herpes virus G47∆ for residual or recurrent glioblastomas	Phase II Single-arm trial UMIN000015995	G47∆ administered intratumorally First and second doses: 5–14-day intervals; Third and subsequent doses: up to six doses at 4 ± 2-week intervals; Dosage: 1 × 10^9^ pfu per dose in a 1 mL solution.	30 patients	Primary endpoint (1-year survival rate): 84.2% (95% CI, 60.4–96.6). Secondary endpoints: median overall survival: 20.2 months after G47∆ initiation; 28.8 months from initial surgery. Enlargement and contrast-enhancement clearing in the target lesion after each G47∆ administration. Increase in tumor-infiltrating CD4+/CD8+ lymphocytes. Persistent low numbers of Foxp3+ cells. Demonstrated survival benefit. Good safety profile. Led to the approval of G47∆ as the first oncolytic virus product in Japan.	Japan	2015–2018	[35]
Combined immunotherapy with controlled interleukin-12 gene therapy and immune checkpoint blockade in recurrent glioblastomas (rGBMs)	Phase I trial, open-label, multi-institutional, dose-escalation NCT03636477	Nivolumab administration: 7 (±3) days before the resection of the rGBM Ongoing every 2 weeks after surgery. Dosages in cohorts: 1 mg/kg; 3 mg/kg. VDX administration: 3 h before surgery Continued for 14 days after surgery. Dosages in cohorts: 10 mg; 20 mg. IL-12 gene therapy: Peritumoral injection after the resection of the rGBM.	21 patients	VDX 10 mg with nivolumab: 16.9 months. For all subjects: 9.8 months. Controlled IL-12 gene therapy with nivolumab was safe in recurrent GBM patients. This combination immunotherapy increased tumor IFNγ, suggesting immune activation. The safety of combining immunotherapy was confirmed, prompting the initiation of a Phase II clinical trial (NCT04006119).	USA	2018–2019	[36]

## Data Availability

Not applicable.

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
