# Peer review of "Advances in Diagnostic Tools and Therapeutic Approaches for Gliomas: A Comprehensive Review"

_sensors, 2023, doi:10.3390/s23249842_

Round 1

Reviewer 1 Report

Comments and Suggestions for Authors

- Some table is needed to be added in discussion. 

- The trends and continuity in the researches should be discussed. 

Author Response

I would like to express my sincere gratitude for your time and thoughtful feedback on our manuscript. Your insightful comments and suggestions have been invaluable in refining the quality and clarity of the work. We appreciate the effort you have invested in reviewing our manuscript and the constructive nature of your comments. Your expertise has significantly contributed to enhancing the overall rigor and impact of the research. We have carefully considered each of your comments and made revisions accordingly. Your dedication to maintaining the scientific integrity of the manuscript is truly commendable.

Reviewer 01

Some table is needed to be added in discussion.

The trends and continuity in the researches should be discussed.

We have included a comprehensive table titled "Review of Clinical Trials for Glioblastoma Treatment”. This table serves to provide readers with a detailed overview of ongoing clinical trials, offering valuable insights into the current landscape of Glioblastoma treatment studies. - line number 188

Reviewer 2 Report

Comments and Suggestions for Authors

This article reviewed the recent advancement of theragnostic sensors to conquer glioma, such as computed tomography (CT), magnetic resonance imaging (MRI), optical coherence tomography, surface-enhanced Raman spectroscopy, and bioelectronics sensors. The disadvantages and advantages of biocompatible nanomaterials, implantable electronics, memristive computing, and the management of cytokine release syndrome were also discussed in this article. This review proposed some perspectives that are expected to be meaningful for next generation sensor design and development for conquering gliomas. Therefore, I recommend its acceptance after minor revision.

(1)   Some typing errors should be carefully checked in the manuscript. For example, in Line 589 on Page 13, a full stop (.) was missed after ‘in vitro’. In Line 694 on Page 15, two full stops were used after ‘model’.

(2) I think the heading 'Introduction' is inappropriate for the part 1. I recommend that sections 1.1-1.5 constitutes part 2 which is given a new heading.

Author Response

I would like to express my sincere gratitude for your time and thoughtful feedback on our manuscript. Your insightful comments and suggestions have been invaluable in refining the quality and clarity of the work. We appreciate the effort you have invested in reviewing our manuscript and the constructive nature of your comments. Your expertise has significantly contributed to enhancing the overall rigor and impact of the research. We have carefully considered each of your comments and made revisions accordingly. Your dedication to maintaining the scientific integrity of the manuscript is truly commendable.

(1)   Some typing errors should be carefully checked in the manuscript. For example, in Line 589 on Page 13, a full stop (.) was missed after ‘in vitro’. In Line 694 on Page 15, two full stops were used after ‘model’.

(2) I think the heading 'Introduction' is inappropriate for the part 1. I recommend that sections 1.1-1.5 constitutes part 2 which is given a new heading.

All typographical errors have been carefully reviewed and rectified throughout the manuscript. Specifically, Sections 1.1 through 1.5 have been renumbered to Sections 2.1 through 2.6, and a new heading has been introduced, labeled "2. Diagnostic Tools." Within this section, the following subheadings have been incorporated:

  1. Diagnostic tools – (line number 200)

2.1 Colorimetric Technique for Brain Cancer Diagnostic: Tumor Markers (line number 201)

2.2 Electrochemical Biosensors for Brain Cancer Diagnostic via Tumor Biomarkers (Line number 323)

2.3 Optical Coherence Tomography  (Line number 402)

2.4 Surface-Enhanced Raman Spectroscopy (Line number 651)

2.5 Reflectometric Interference Spectroscopy (Line number 868)

2.6 Optical Biosensors (Line number 994)

Reviewer 3 Report

Comments and Suggestions for Authors

1)      The authors should revise the abstract. It is quite long abstract, but it is not in order well. It should start with glioma, and then problems with current diagnostic approaches. After that authors should describe the aim of the review and briefly describe some latest results to diagnose glioma with a focus on numerical results.

2)      I believe the figure 1 is not a graphical abstract. It describes (as mentioned in the figure’s legend) the challenges in glioma research. The authors require a graphical abstract as Figure 1 and the present Figure 2 could be moved down. (somewhere in line 180)

3)      In sections 1.1-1.5, the authors described various biosensors based on signal detectors (optical, colorimetric …), however, they did not discuss electrochemical biosensors for brain cancer diagnostic via tumor biomarkers.  Electrochemical biosensors are an ideal technique for those biomarkers that are very low amount ( aM-zM). The authors could use this recently published reference.

https://www.sciencedirect.com/science/article/pii/S2666831923000929

4)      It would be nice if the authors mentioned any available technologies presented in the 1.1-1.5 sections for the diagnosis of brain cancer.

5)      I am wondering about “ theragnostic sensors” in the title of the manuscript. A sensor is a device to senses the analyte. The authors also used diagnostic sensors in 1.1-1.5 and 2.1-2.3 sections. I do not understand why they used “ theragnostic sensor” They may have to change it.

6)      The authors described microchips in section 3, the drug delivery system (in section 3.1), and gene therapy (in section 3.2). Is there any available or under R&D technology? I should mention these are Theragnostic chips or devices, not a sensor.

7)  The quality of Figures 2,4, and 5 is very poor. They must be revised.

Author Response

Response to Reviewer 03

I would like to express my sincere gratitude for your time and thoughtful feedback on our manuscript. Your insightful comments and suggestions have been invaluable in refining the quality and clarity of the work. We appreciate the effort you have invested in reviewing our manuscript and the constructive nature of your comments. Your expertise has significantly contributed to enhancing the overall rigor and impact of the research. We have carefully considered each of your comments and made revisions accordingly. Your dedication to maintaining the scientific integrity of the manuscript is truly commendable.

  1. The authors should revise the abstract. It is quite long abstract, but it is not in order well. It should start with glioma, and then problems with current diagnostic approaches. After that authors should describe the aim of the review and briefly describe some latest results to diagnose glioma with a focus on numerical results.

The abstract has been revised accordingly (Line 14-40).

  1. In sections 1.1-1.5, the authors described various biosensors based on signal detectors (optical, colorimetric …), however, they did not discuss electrochemical biosensors for brain cancer diagnostic via tumor biomarkers.  Electrochemical biosensors are an ideal technique for those biomarkers that are very low amount (aM-zM).

The section on Electrochemical Biosensors has been incorporated into Section 2, titled "Diagnostic Tools (Line 200)," specifically as 2.1 Electrochemical Biosensors for Brain Cancer Diagnostics via Tumor Biomarkers (Line 201).

  1. I am wondering about “theragnostic sensors” in the title of the manuscript. A sensor is a device to senses the analyte. The authors also used diagnostic sensors in 1.1-1.5 and 2.1-2.3 sections. I do not understand why they used “ theragnostic sensor” They may have to change it.

The title of the manuscript has been revised to "Advances in Diagnostic Tools and Therapeutic Approaches for Glioma: A Comprehensive Review." (Line 03)

  1. I am wondering about “theragnostic sensors” in the title of the manuscript. A sensor is a device to senses the analyte. The authors also used diagnostic sensors in 1.1-1.5 and 2.1-2.3 sections. I do not understand why they used “theragnostic sensor” They may have to change it.

‘Theragnostic’ has been removed in the manuscript.

  1. The authors described microchips in section 3, the drug delivery system (in section 3.1), and gene therapy (in section 3.2). Is there any available or under R&D technology? I should mention these are Theragnostic chips or devices, not a sensor.

The concept has been corrected through the text.

  1. The quality of Figures 2,4, and 5 is very poor. They must be revised.

The structure of the review has been revised to enhance clarity and organization, with figures 2, 4, and 5 now designated and revised as figure 3 (Line 258), 4 (Line 1185), and 5 (Line 1357), respectively. The updated structure is outlined as follows:

  1. Introduction (Line 47)
  1. Diagnostic Tools (Line 200)

2.1 Colorimetric Technique for Brain Cancer Diagnostic: Tumor Markers (line number 201)

2.2 Electrochemical Biosensors for Brain Cancer Diagnostic via Tumor Biomarkers (Line number 323)

  • Optical Coherence Tomography (Line number 402)

2.4 Surface-Enhanced Raman Spectroscopy (Line number 651)

2.5 Reflectometric Interference Spectroscopy (Line number 868)

2.6 Optical Biosensors (Line number 994)Biotechnology Tools (Line 1143)

3.1 Drug Delivery System (Line 1154)

3.2 Gene Circuits (Line 1190)

3.3 Fcγ CR T Cell Immunotherapy (Line 1312)

3.4 Targeted Treatment, Cytokine Release Syndrome Management, and Advanced Nanoplatform Systems (Line 1384)

  1. Bioelectronic Sensors (Line 1410)

4.1 Nanomaterials as an Interface for Targeting Glioma (Line 1422)

4.2 Ultraminiaturized, Wirelessly Charged, and Biocompatible Implantable Electronics (Line 1492)

4.3 Neuromorphic and Memristive Computing (Line 1543)

4.4 Multimodal, Multi-Site, and Adaptive In-Brain Glioma Therapeutics (Line 1650)

4.5 Microchip Hardware Development 4.6 Cytokine Sensors Development (Line 1676)

  1. Conclusions (Line 1707)

Reviewer 4 Report

Comments and Suggestions for Authors

I appreciate the authors' efforts in addressing an important topic in glioma research. However, I have identified several areas that require attention and improvement before the manuscript can be considered for publication.

1.     The current title, "Development of Theragnostic Sensors to Conquer Glioma," does not accurately reflect the content of the article. It implies a focus on the development of sensors, which is not the primary emphasis of the manuscript. I suggest revising the title to better align with the content. A more appropriate title might be "Advances in Diagnostic Tools and Therapeutic Approaches for Glioma: A Comprehensive Review."

2.     The introduction is vague and lacks clarity. It should provide a concise overview of the current understanding of glioma, recent molecular classification, and advancements in clinical trials and gene therapy. The introductory section should also set the context for the rest of the review. I recommend rewriting the introduction to incorporate these elements and provide a clear roadmap for the reader.

3.     Manuscript lacks recent citations related to the molecular classification of gliomas and recent advancements in clinical trials and gene therapy. It is essential to incorporate up-to-date references to provide a comprehensive and current overview of the field. Some relevant studies to consider include those by Todo et al. (Japan), Chiocca et al. (USA), and Ummeura et al. (USA), among others.

4.     Manuscript's content appears scattered and could benefit from better organization. Each subsection should flow logically from the previous one, creating a cohesive narrative for the reader. It would be helpful to create a structured outline and ensure that each section contributes to the overall understanding of glioma diagnosis and treatment.

5.     The manuscript contains several lengthy sentences that could be broken down for improved readability. Additionally, some sentences and paragraphs are overly complex and could benefit from simplification.

6.     Consider including tables to illustrate data, particularly in sections discussing diagnostic tools and therapeutic approaches.

7.     The conclusion should summarize the key findings and insights from the review and highlight the significance of the discussed advancements in glioma research. Ensure that the conclusion aligns with the main objectives of the manuscript.

Comments on the Quality of English Language

Furthermore, it is essential to address the quality of English language usage throughout the manuscript. While the content is valuable, there are instances of awkward phrasing and grammatical errors that need correction. I recommend a thorough proofreading and language editing to ensure that the manuscript reads smoothly and is free from language-related distractions.

Author Response

I would like to express my sincere gratitude for your time and thoughtful feedback on our manuscript. Your insightful comments and suggestions have been invaluable in refining the quality and clarity of the work. We appreciate the effort you have invested in reviewing our manuscript and the constructive nature of your comments. Your expertise has significantly contributed to enhancing the overall rigor and impact of the research. We have carefully considered each of your comments and made revisions accordingly. Your dedication to maintaining the scientific integrity of the manuscript is truly commendable.

  1. The current title, "Development of Theragnostic Sensors to Conquer Glioma," does not accurately reflect the content of the article. It implies a focus on the development of sensors, which is not the primary emphasis of the manuscript. I suggest revising the title to better align with the content. A more appropriate title might be "Advances in Diagnostic Tools and Therapeutic Approaches for Glioma: A Comprehensive Review."

The title of the abstract has been revised to "Advances in Diagnostic Tools and Therapeutic Approaches for Glioma: A Comprehensive Review."

  1. The introduction is vague and lacks clarity. It should provide a concise overview of the current understanding of glioma, recent molecular classification, and advancements in clinical trials and gene therapy. The introductory section should also set the context for the rest of the review. I recommend rewriting the introduction to incorporate these elements and provide a clear roadmap for the reader.

Introduction has been rewritten and provided a clear roadmap for the reader.

  1. Manuscript lacks recent citations related to the molecular classification of gliomas and recent advancements in clinical trials and gene therapy. It is essential to incorporate up-to-date references to provide a comprehensive and current overview of the field. Some relevant studies to consider include those by Todo et al. (Japan), Chiocca et al. (USA), and Ummeura et al. (USA), among others.

The introduction has been enhanced with recent molecular classifications and a concise summary of recent clinical trials. Notably, we have included a comprehensive table titled "Review of Clinical Trials for Glioblastoma Treatment." This table aims to provide readers with a detailed overview of ongoing clinical trials, offering valuable insights into the current landscape of Glioblastoma treatment studies. Additionally, we have updated references to ensure the content is current and reflective of the latest advancements in the field, contributing to a thorough and up-to-date understanding.

Manuscript's content appears scattered and could benefit from better organization. Each subsection should flow logically from the previous one, creating a cohesive narrative for the reader. It would be helpful to create a structured outline and ensure that each section contributes to the overall understanding of glioma diagnosis and treatment.

The structure of the review has been revised to enhance clarity and organization, with figures 2, 4, and 5 now designated and revised as figure 3 (Line 258), 4 (Line 1185), and 5 (Line 1357), respectively. The updated structure is outlined as follows:

  1. Introduction (Line 47)
  1. Diagnostic Tools (Line 200)

2.1 Colorimetric Technique for Brain Cancer Diagnostic: Tumor Markers (line number 201)

2.2 Electrochemical Biosensors for Brain Cancer Diagnostic via Tumor Biomarkers (Line number 323)

  • Optical Coherence Tomography (Line number 402)

2.4 Surface-Enhanced Raman Spectroscopy (Line number 651)

2.5 Reflectometric Interference Spectroscopy (Line number 868)

2.6 Optical Biosensors (Line number 994)Biotechnology Tools (Line 1143)

3.1 Drug Delivery System (Line 1154)

3.2 Gene Circuits (Line 1190)

3.3 Fcγ CR T Cell Immunotherapy (Line 1312)

3.4 Targeted Treatment, Cytokine Release Syndrome Management, and Advanced Nanoplatform Systems (Line 1384)

  1. Bioelectronic Sensors (Line 1410)

4.1 Nanomaterials as an Interface for Targeting Glioma (Line 1422)

4.2 Ultraminiaturized, Wirelessly Charged, and Biocompatible Implantable Electronics (Line 1492)

4.3 Neuromorphic and Memristive Computing (Line 1543)

4.4 Multimodal, Multi-Site, and Adaptive In-Brain Glioma Therapeutics (Line 1629)

4.5 Microchip Hardware Development (Line 1650)

4.6 Cytokine Sensors Development (Line 1676)

  1. Conclusions (Line 1707)

  1. The manuscript contains several lengthy sentences that could be broken down for improved readability. Additionally, some sentences and paragraphs are overly complex and could benefit from simplification.

 Sentences have been shorted.

  1. Consider including tables to illustrate data, particularly in sections discussing diagnostic tools and therapeutic approaches.

 We have included a comprehensive table titled "Review of Clinical Trials for Glioblastoma Treatment”. This table serves to provide readers with a detailed overview of ongoing clinical trials, offering valuable insights into the current landscape of Glioblastoma treatment studies. - line number 188

  1. The conclusion should summarize the key findings and insights from the review and highlight the significance of the discussed advancements in glioma research. Ensure that the conclusion aligns with the main objectives of the manuscript.

Conclusion has been rewritten.

 Comments on the Quality of English Language

Furthermore, it is essential to address the quality of English language usage throughout the manuscript. While the content is valuable, there are instances of awkward phrasing and grammatical errors that need correction. I recommend a thorough proofreading and language editing to ensure that the manuscript reads smoothly and is free from language-related distractions.

Round 2

Reviewer 3 Report

Comments and Suggestions for Authors

The authors answered my comment and carried out the correction, therefore I recommend publishing.